# FEDERATED SKETCHING LoRA: A FLEXIBLE FRAMEWORK FOR HETEROGENEOUS COLLABORATIVE FINE-TUNING OF LLMS

## ABSTRACT

Fine-tuning large language models (LLMs) on resource-constrained clients remains a challenging problem. Recent works have fused low-rank adaptation (LoRA) techniques with federated fine-tuning to mitigate challenges associated with client model sizes and data scarcity. Still, the heterogeneity of resources remains a critical bottleneck: while higher-rank modules generally enhance performance, varying client capabilities constrain LoRA's feasible rank range. Existing approaches attempting to resolve this issue either lack analytical justification or impose additional computational overhead, leaving a wide gap for efficient and theoretically-grounded solutions. To address these challenges, we propose federated sketching LoRA (FSLoRA), which leverages a sketching mechanism to enable clients to selectively update submatrices of global LoRA modules maintained by the server. By adjusting the sketching ratios, which determine the ranks of the submatrices on the clients, FSLoRA flexibly adapts to client-specific communication and computational constraints. We provide a rigorous convergence analysis of FSLoRA that characterizes how the sketching ratios affect the convergence rate. Through comprehensive experiments on multiple datasets and LLM models, we demonstrate FSLoRA's performance improvements compared to various baselines.

## 1 INTRODUCTION

Lightweight client-side large language models (LLMs) have recently gained significant attention as a promising complement to cloud-based LLMs (Fan et al., 2024). They align with the typical paradigm of LLMs: starting from a base model pre-trained on large-scale datasets to learn general linguistic patterns, semantics, and context, and then undergoing fine-tuning on task-specific data to enhance performance on specialized or domain-specific applications. However, an LLM fine-tuned on a single client often achieves unsatisfactory performance due to the limited data. Federated learning (McMahan et al., 2017; Chen et al., 2023) has been investigated as a potential solution here, enabling the model to be fine-tuned across a group of distributed clients within the same task domain, without any raw data sharing.

However, federated LLM fine-tuning is costly in both computation and communication due to the massive parameter volume. Importantly, many parameter-efficient fine-tuning methods have been proposed (Lester et al., 2021; Li and Liang, 2021; Hu et al., 2021) to reduce the model adaptation cost. Among them, low-rank adaptation (LoRA) (Hu et al., 2021) stands out as a particularly effective approach due to its flexibility. In particular, LoRA enables efficient fine-tuning by approximating weight updates $\Delta\mathbf{W}$ through a low-rank decomposition $\Delta\mathbf{W} = \mathbf{BA}$, where matrices $\mathbf{B}$ and $\mathbf{A}$ contain significantly fewer trainable parameters than the original weight matrix. Building on this foundation, recent works have combined LoRA with federated averaging (FedAvg) (Zhang et al., 2024; Ye et al., 2024), showing that federated LoRA significantly reduce the training overhead.

**Challenges.** While incorporating LoRA into federated LLM fine-tuning reduces the number of trainable parameters, *computation and communication costs are still forced to increase with the LoRA rank*. This poses challenges when complex tasks demand higher-rank LoRA modules, particularly on resource-constrained clients. Furthermore, the *heterogeneity in resource availability across distributed clients makes a uniform rank adopted in federated LoRA inefficient*: a fixed rank $r$ may be too large

for some constrained clients, while being too small for more powerful ones, resulting in underutilized resources. Consequently, an approach that further reduces computation and communication overhead while adapting LoRA ranks to heterogeneous client capabilities is highly desirable. Although some existing approaches have attempted to provide a solution (Cho et al., 2024; Bai et al., 2024; Wang et al., 2024), they either lack theoretical justification or impose additional computational overhead, leaving a gap for an efficient and theoretically-grounded solution. As we discuss in Section 2.2, a comprehensive approach that preserves the analytical and practical benefits of LoRA while enabling heterogeneous collaborative fine-tuning under tight resource constraints remains elusive.

## 1.1 CONTRIBUTIONS

Motivated by these limitations, this work develops a methodology for efficient federated LLM fine-tuning that (i) retains the flexibility of LoRA, (ii) provides theoretical convergence guarantees, and (iii) addresses the challenges posed by heterogeneous and constrained resources across distributed clients. As depicted in Figure 1, our key idea is to introduce a sketching-based LoRA update to the local fine-tuning, which allows clients to selectively update a subset of columns and rows of the LoRA modules during each round, reducing the computation and communication consumption. Additionally, our method customizes the fine-tuning process by adjusting the sparsity level of the sketching matrix, i.e., the size of the updated submatrices for

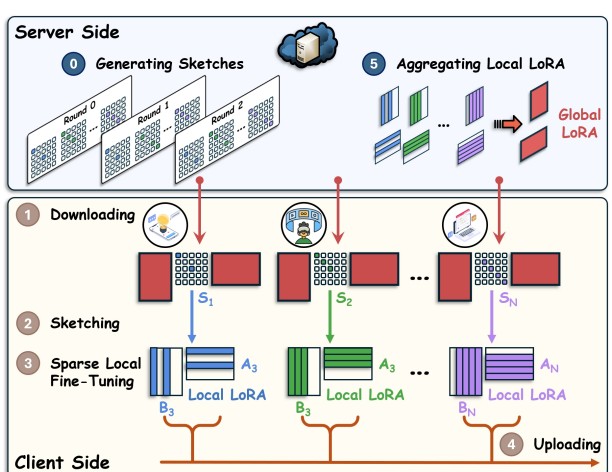

Figure 1: An illustration of our proposed methodology where the server maintains a pair of global LoRA modules while the clients adaptively update submatrices of the global LoRA modules through sketching during each round.

each client in each iteration. As we will see, the impact of the introduced sketching mechanism on the overall optimization landscape requires careful modeling consideration, posing additional challenges for the theoretical analysis that we address in this work.

Overall, we make the following contributions:

- We propose federated sketching LoRA (FSLoRA), which leverages a sketching mechanism to enable clients to selectively update submatrices of global LoRA modules maintained by the server. By adjusting the sketching ratios, which determine the ranks of the submatrices on clients, FSLoRA effectively adapts to client-specific communication and computational constraints.

- We present a rigorous convergence analysis of FSLoRA under non-uniform submatrix update scenarios (i.e., heterogeneous LoRA configurations) across clients, revealing how the sketching ratios affect the convergence rate via scaled smoothness constants. Further, our results show that while increasing the sketching ratios improves convergence theoretically, it also raises communication and computation costs, suggesting a potential trade-off in selecting the sketching ratios.

- We conduct extensive experiments across multiple datasets and LLM models with diverse parameter settings, demonstrating FSLoRA's superior performance compared to various baselines in accuracy, training time, and resource utilization. Our ablation studies further validate the effectiveness of the sketching mechanism and the ability of clients to exploit larger global ranks under FSLoRA.

## 1.2 RELATED WORKS

**Collaborative fine-tuning via federated LoRA:** Federated LoRA is an efficient approach for collaborative LLM fine-tuning among distributed clients (Chen et al., 2023; Sun et al., 2024; Guo et al., 2025). Building on this foundation, Kuo et al. (2024) proposed integrating communication compression with federated LoRA to further reduce communication overhead. Meanwhile, Bai et al. (2024); Cho et al. (2024); Byun and Lee (2024); Wang et al. (2024); Koo et al. (2024) explored the challenges of resource heterogeneity across distributed clients and introduced heterogeneous LoRA as a solution. However, the approaches proposed in (Cho et al., 2024; Koo et al., 2024; Byun and Lee,

2024; Bai et al., 2024) lack a theoretical foundation. Moreover, the FlexLoRA method introduced in (Bai et al., 2024) incurs additional computational overhead due to its reliance on singular value decomposition (SVD). Furthermore, the FLoRA algorithm proposed in (Wang et al., 2024) requires the clients to merge the LoRA modules into the base model, thereby compromising the inherent flexibility of LoRA. Overall, there is still a lack of a systematic and theoretically grounded solution that can effectively tackle heterogeneous collaborative LLM fine-tuning.

**Enhancing adaptability via higher-rank LoRA modules:** The foundational study by Hu et al. (2021) demonstrated that small ranks can be sufficient for certain tasks; however, they also acknowledge that small rank LoRA modules may not work universally, especially when the downstream task differs significantly from pretraining. Following this, several works explored the effect of increasing the rank in LoRA modules. In a centralized setup, Kalajdzievski (2023) and Shuttleworth et al. (2024) showed that higher-rank LoRA models can closely approximate full fine-tuning under rsLoRA. In a federated LLM fine-tuning regime, Bai et al. (2024) demonstrated improved performance with larger ranks under FlexLoRA. Similarly, Cho et al. (2024) reported that, with proper overfitting control, HeteroLoRA can also benefit from larger ranks. Overall, while small ranks may suffice for simpler tasks or strong base models, higher-rank modules are necessary to compensate for limited backbone capability, such as in lightweight LLMs, and to enable effective adaptation to more complex downstream tasks.

**Sketching-based optimization:** Sketching is an efficient technique for mitigating the complexity of high-dimensional optimization, with its earliest applications in least-squares regression (Sarlos, 2006; Wang et al., 2022). Beyond this, gradient sketching has been employed to construct preconditioners for gradient descent methods (Gower and Richtárik, 2015). Building on these foundations, recent work has applied sketching to distributed optimization. In particular, Charalambides and Mazumdar (2024) proposed hybrid local-global sketching for distributed least-squares, while Demidovich et al. (2023) developed a distributed sparsified training framework based on sketching. Shrivastava et al. (2024) demonstrated that sketching substantially reduces communication in distributed training of overparameterized deep models without sacrificing accuracy. More recently, Nicolas et al. (2025) investigated sketching-based differential privacy and demonstrated its compatibility with secure aggregation. Despite these advances, sketching strategies tailored to structured low-rank adaptation modules such as LoRA remain largely unexplored.

## 2 PROBLEM BACKGROUND

### 2.1 LoRA-BASED FEDERATED LLM FINE-TUNING

The federated LoRA fine-tuning problem can be formulated as

$$\min_{\mathbf{B},\mathbf{A}} f(\mathbf{B},\mathbf{A}) \coloneqq \frac{1}{N}\sum_{i=1}^{N} f_i(\mathbf{B},\mathbf{A}), \text{ where } f_i(\mathbf{B},\mathbf{A}) \coloneqq \mathbb{E}_{\xi\sim\mathcal{D}_i}\left[\ell(\mathbf{W}_0 + \mathbf{B}\mathbf{A}, \xi)\right], \quad (1)$$

where $\mathbf{W}_0$ denotes the frozen base model, $\mathbf{B}\in\mathbb{R}^{m\times r}$, $\mathbf{A}\in\mathbb{R}^{r\times n}$ are LoRA modules, $N$ denotes the number of clients, $\xi$ denotes a data sample, and $\mathcal{D}_i$ is the local dataset on client $i$. $\ell$, $f_i$, and $f$ are the sample loss function, the local loss for client $i$, and the global loss, respectively.

Problem (1) aligns with the conventional federated optimization formulation, which thus can be solved using the FedAvg algorithm. Based on the FedAvg framework, Zhang et al. (2024) developed federated LoRA, which applies a uniform rank $r$ across all clients, overlooking resource heterogeneity. This one-size-fits-all approach leads to resource mismatches, where computationally constrained clients may struggle, while more powerful clients remain underutilized with a fixed rank.

### 2.2 AREN'T THE EXISTING SOLUTIONS GOOD ENOUGH?

To address this issue, researchers have proposed heterogeneous federated LoRA approaches, where clients maintain non-uniform LoRA modules with varying ranks. They also introduce mechanisms to overcome the challenges of directly aggregating matrices with different dimensions. However, these methods often lack theoretical foundation or incur additional computational and memory overhead.

**HeteroLoRA (Cho et al., 2024)** lets the server pad the updates from the clients with smaller ranks to match the size of the largest rank during aggregation. During model dissemination, clients receive a truncated version of the global LoRA modules from the server. Although easy to implement,

HeteroLoRA is primarily heuristic in nature and lacks a rigorous theoretical foundation, potentially limiting its performance, as we will see in Section 5.

**FlexLoRA (Bai et al., 2024)** requires the server to collect the individual LoRA matrices $\mathbf{B}_i$ and $\mathbf{A}_i$ from the clients and then computes their product $\mathbf{B}_i\mathbf{A}_i$. To support the initialization of non-uniform LoRA modules, the server applies truncated SVD to the averaged product $\frac{1}{N}\sum_{i=1}^{N}\mathbf{B}_i\mathbf{A}_i$. However, this approach introduces extra computational and memory overhead on the server due to truncated SVD, and the associated error can limit the performance as demonstrated in Section 5.

**FLoRA (Wang et al., 2024)** introduces a stacking mechanism where the server concatenates LoRA modules from the clients. The concatenated matrices are then sent back to the clients, which compute their product and merge it into the base model before initializing new LoRA modules for the next fine-tuning round. However, this approach increases communication complexity linearly with the number of clients, imposes higher computation and memory demands on the clients, and compromises LoRA's flexibility to support multiple adapters for different tasks.

More detailed comparisons on computation, memory, and communication are presented in Appendix A. In summary, a theoretically-grounded solution that preserves LoRA's benefits while effectively addressing resource heterogeneity across distributed clients remains lacking.

## 3 FEDERATED SKETCHING LoRA

Motivated by the limitations of existing methods, we propose a new federated LoRA reformulation. Building on this foundation, we develop FSLoRA, a heterogeneous LoRA algorithm that preserves LoRA's flexibility while accommodating client resource heterogeneity.

### 3.1 OUR FORMULATION

We propose a sketching-based LoRA formulation for collaborative LLM fine-tuning as follows:

$$\min_{\mathbf{B},\mathbf{A}} f^{\mathcal{S}}(\mathbf{B},\mathbf{A}) \coloneqq \frac{1}{N}\sum_{i=1}^{N} f_i^{\mathcal{S}}(\mathbf{B},\mathbf{A}) \text{ where } f_i^{\mathcal{S}}(\mathbf{B},\mathbf{A}) \coloneqq \mathbb{E}_{\mathbf{S}\sim\mathcal{S}_i;\xi\sim\mathcal{D}_i}\left[\ell(\mathbf{W}_0 + \mathbf{B}\mathbf{S}\mathbf{A},\xi)\right], \quad (2)$$

where $\mathbf{B}\in\mathbb{R}^{m\times r}, \mathbf{A}\in\mathbb{R}^{r\times n}$ are LoRA modules, $f_i^{\mathcal{S}}$ is the local loss function at client $i$ with sketching, and $\mathbf{S}$ denotes a sketching matrix randomly sampled from the diagonal matrix set $\mathcal{S}_i = \mathcal{S}(r,k_i)$. The set $\mathcal{S}(r,k_i)$ comprises diagonal matrices of size $r\times r$ with exactly $k_i$ non-zero entries. The formal definition of $\mathcal{S}(r,k)$ is provided below:

**Definition 3.1** (Random-$k$ sketching). A random-$k$ diagonal matrix set is defined as:

$$\mathcal{S}(r,k) = \left\{\mathbf{S}\,|\,\mathbf{S} = \frac{r}{k}\sum_{j\in\mathcal{I}}\mathbf{e}_j\mathbf{e}_j^\top, \mathcal{I}\subseteq\{1,\ldots,r\}, |\mathcal{I}| = k\right\}, \quad (3)$$

where $\mathbf{e}_1,\ldots,\mathbf{e}_r\in\mathbb{R}^r$ are standard unit basis vectors and index set $\mathcal{I}$ is a random subset of $[r]\coloneqq\{1,2,\ldots,r\}$ sampled uniformly from all subsets of $[r]$ with cardinality $k$.

With $\mathbf{S}$ being a matrix sampled from $\mathcal{S}_i$, we have $\mathbf{B}\mathbf{S}\mathbf{A} = \frac{r}{k_i}\sum_{j\in\mathcal{I}_i}\mathbf{B}\mathbf{e}_j\mathbf{e}_j^\top\mathbf{A}$, where $\mathcal{I}_i$ corresponds to the index set of non-zero diagonal entries of $\mathbf{S}$. $\mathbf{B}\mathbf{e}_j$ extracts the $j$-th column of $\mathbf{B}$ while $\mathbf{e}_j^\top\mathbf{A}$ extracts the $j$-th row of $\mathbf{A}$. In other words, only $k_i$ columns and rows in the LoRA modules $\mathbf{B}$ and $\mathbf{A}$ are activated by the sketching matrix in the loss $\ell(\mathbf{W}_0 + \mathbf{B}\mathbf{S}\mathbf{A},\xi)$ at client $i$. On the other hand, the sketching matrix $\mathbf{S}$ satisfies $\mathbb{E}_{\mathbf{S}\sim\mathcal{S}_i}[\mathbf{S}] = \mathbf{I}_r$ where $\mathbf{I}_r$ is a $r$-dimensional identity matrix. Based upon this property, $\mathbf{W}_0 + \mathbf{B}\mathbf{S}\mathbf{A}$ can be treated as an unbiased estimate of $\mathbf{W}_0 + \mathbf{B}\mathbf{A}$.

**Intuition:** A larger rank allows LoRA modules to be more expressive, leading to better performance (Bai et al., 2024; Kalajdzievski, 2023; Shuttleworth et al., 2024). However, resource-constrained clients cannot afford the computational or communication demands of large-rank modules. Our formulation (2) leverages the sketching matrix to balance the expressiveness of high-rank LoRA modules with the resource constraints of different clients. With the sketching mechanism introduced, the local gradients with respect to the LoRA modules on the clients will exhibit structured sparsity. By adjusting the sketching ratio $k_i/r$, we can tailor the sparsity of the gradient to match the capabilities of each client, ensuring affordable training while maintaining performance across heterogeneous systems, as elaborated in the following subsection. Overall, compared to (1), our formulation offers a more flexible framework, tailored to address the diverse capabilities of heterogeneous clients.

### 3.2 SPARSITY IN THE GRADIENTS

In this subsection, we analyze the gradient structure of LoRA modules and highlight the gradients' sparsity properties under a given sketching matrix. To begin, we present the gradient expressions for the LoRA modules $\mathbf{B}$ and $\mathbf{A}$ in the following lemma. The proof is provided in Appendix J.2.

**Lemma 3.2** (Gradient Formulation). *For a given sketching matrix $\mathbf{S}$, the gradients of $\ell(\mathbf{W}_0 + \mathbf{BSA}, \xi)$ with respect to $\mathbf{B}$ and $\mathbf{A}$ take the following form*

$$\nabla_{\mathbf{B}}\ell(\mathbf{W}_0 + \mathbf{BSA}, \xi) = \nabla\ell(\mathbf{W}_0 + \mathbf{BSA}, \xi)\mathbf{A}^{\top}\mathbf{S}^{\top}$$
$$\nabla_{\mathbf{A}}\ell(\mathbf{W}_0 + \mathbf{BSA}, \xi) = \mathbf{S}^{\top}\mathbf{B}^{\top}\nabla\ell(\mathbf{W_0} + \mathbf{BSA}, \xi),$$
(4)

*where $\nabla_{\mathbf{B}}\ell(\mathbf{W}_0 + \mathbf{BSA}, \xi)$, $\nabla_{\mathbf{A}}\ell(\mathbf{W}_0 + \mathbf{BSA}, \xi)$, and $\nabla\ell(\mathbf{W}_0 + \mathbf{BSA}, \xi)$ represent the gradients of $\ell(\mathbf{W}_0 + \mathbf{BSA}, \xi)$ with respect to $\mathbf{B}$, $\mathbf{A}$, and $\mathbf{W_0} + \mathbf{BSA}$, respectively.*

In particular, a random-$k$ diagonal sketching matrix selectively samples $k$ rows or columns of a matrix through left product or right product, respectively. With $\mathbf{S}$ being a random-$k$ diagonal matrix, the gradients of $\ell(\mathbf{W}_0 + \mathbf{BSA}, \xi)$ with respect to LoRA modules $\mathbf{B}$ and $\mathbf{A}$, as shown in (4), naturally become structurally sparse matrices. This sparsity reduces computational and memory overhead during training, enabling faster gradient computation and parameter updates, while alleviating communication overhead across distributed clients by transmitting only non-zero elements.

**Remark 3.3** (Sparsity Level Control). A key advantage of our formulation is its flexible control over the sparsity level of local gradients, achieved by configuring the parameter $k_i$ of the sketching matrix set $\mathcal{S}_i = \mathcal{S}(r, k_i)$. This mechanism allows each client to tailor its local updates according to its communication and computation resource constraints, ensuring efficient and scalable fine-tuning in heterogeneous federated systems. Lowering $k_i$ helps resource-constrained clients reduce computation and communication overhead, while more capable clients can increase $k_i$ to conduct more informative local updates. Additionally, the distinction in sparsity level control between the proposed FSLoRA and the FedBCGD algorithm (Liu et al., 2024) is elaborated in Appendix B.

**Remark 3.4** (Justification for the Choice of Random-$k$ Sketching). We adopt Random-$k$ sketching due to its unbiasedness and the structured sparsity it induces. A detailed discussion and empirical comparison with alternative sketching strategies are provided in Appendix C.

### 3.3 FSLoRA ALGORITHM

Based on the formulation in (2), we propose a resource-adaptive algorithm termed FSLoRA for collaborative LLM fine-tuning. FSLoRA allows each client to update submatrices of the original modules $\mathbf{B}$ and $\mathbf{A}$ in each round. The server maintains a pair of global LoRA modules $\mathbf{B}$ and $\mathbf{A}$ and periodically updates them by aggregating sparse local updates received from distributed clients. Specifically, the procedure of FSLoRA at each round is detailed below.

- The server begins by sampling sketching matrices $\{\mathbf{S}_i^t \sim \mathcal{S}_i\}_{i=1}^{N}$ for all clients, where $\mathcal{S}_i$ represents the set of possible sketching matrices for client $i$. These sketches are then sent to the corresponding clients. Additionally, the server broadcasts the current global LoRA modules $[\mathbf{B}^t; \mathbf{A}^t]$ to all clients. Note that the communication load introduced by transmitting the sketching matrix is negligible compared to global LoRA modules, as it involves only *binary sketching indices* (i.e., the diagonal elements of the sketching matrix); see Appendix A for details.

- Clients perform local fine-tuning using sketch $\mathbf{S}_i^t$. Specifically, guided by sketching matrix $\mathbf{S}_i^t$, the update at client $i$ during the $h$-th iteration of the $t$-th round is given by:

$$\left[\mathbf{B}_i^{t,h+1}; \mathbf{A}_i^{t,h+1}\right] = \left[\mathbf{B}_i^{t,h}; \mathbf{A}_i^{t,h}\right] - \gamma\left[\Delta\mathbf{B}_i^{t,h}(\mathbf{S}_i^t)^{\top}; (\mathbf{S}_i^t)^{\top}\Delta\mathbf{A}_i^{t,h}\right],$$
(5)

where $\gamma$ denotes the learning rate and $[\Delta\mathbf{B}_i^{t,h}; \Delta\mathbf{A}_i^{t,h}]$ is a shorthand representation for:

$$\left[\Delta\mathbf{B}_i^{t,h}; \Delta\mathbf{A}_i^{t,h}\right] = \left[\nabla\ell(\mathbf{W}_0 + \mathbf{B}_i^{t,h}\mathbf{S}_i^t\mathbf{A}_i^{t,h}, \xi_i^{t,h})(\mathbf{A}_i^{t,h})^{\top}; (\mathbf{B}_i^{t,h})^{\top}\nabla\ell(\mathbf{W}_0 + \mathbf{B}_i^{t,h}\mathbf{S}_i^t\mathbf{A}_i^{t,h}, \xi_i^{t,h})\right].$$

The update direction in (5) corresponds to the negative stochastic gradient of $\ell(\mathbf{W}_0 + \mathbf{BSA}, \xi)$ with respect to $[\mathbf{B}; \mathbf{A}]$ for a given sketch $\mathbf{S}_i^t$, as established in Lemma 3.2. The total update for client $i$ during one round of training, consisting of $H$ local steps, can be expressed as follows:

$$\left[\mathbf{B}_i^{t,H} - \mathbf{B}_i^{t,0}; \mathbf{A}_i^{t,H} - \mathbf{A}_i^{t,0}\right] = \left[\gamma\left(\sum_{h=0}^{H-1}\Delta\mathbf{B}_i^{t,h}\right)(\mathbf{S}_i^t)^{\top}; \gamma(\mathbf{S}_i^t)^{\top}\left(\sum_{h=0}^{H-1}\Delta\mathbf{A}_i^{t,h}\right)\right].$$
(6)

---

**Algorithm 1** Federated Sketching LoRA (FSLoRA)

---

**Require:** Base model $\mathbf{W}_0$, LoRA modules $\mathbf{B}_0$ and $\mathbf{A}_0$, learning rate $\gamma$, and sketching set $\{\mathcal{S}_i\}_{i=1}^N$
 1: **for** $t = 0, 1, \ldots, T-1$ **do**
 2:  Server samples sketching matrices $\{\mathbf{S}_i^t \sim \mathcal{S}_i\}_{i=1}^N$ and sends them back to the clients
 3:  Server broadcasts the current global LoRA modules to the clients
 4:  **for** $h = 0, 1, \ldots, H-1$ **do**
 5:   Clients update the local LoRA modules via (5)
 6:  **end for**
 7:  Clients upload the non-zero columns of $(\mathbf{B}_i^{t,H} - \mathbf{B}_i^{t,0})$ and the non-zero rows $(\mathbf{A}_i^{t,H} - \mathbf{A}_i^{t,0})$
 8:  Server updates the global LoRA modules via (7)
 9: **end for**

---

From the above equation, we see that only the columns of $\mathbf{B}$ and the rows of $\mathbf{A}$ corresponding to the nonzero entries of $\mathbf{S}_i^t$ are updated during the $t$-th round at client $i$. In essence, $\mathbf{S}_i^t$ selectively activates specific columns of $\mathbf{B}$ and rows of $\mathbf{A}$ for each round. Afterward, clients transmit these nonzero columns and rows of the sparse model updates to the server.

- Using the sketch information, the server reconstructs the corresponding sparse matrices from the received updates and aggregates them to update the global model:

$$\left[\mathbf{B}^{t+1}; \mathbf{A}^{t+1}\right] = \left[\mathbf{B}^t; \mathbf{A}^t\right] + \frac{1}{N} \sum_{i=1}^N \left[\mathbf{B}_i^{t,H} - \mathbf{B}_i^{t,0}; \mathbf{A}_i^{t,H} - \mathbf{A}_i^{t,0}\right]. \tag{7}$$

Over training, random-$k$ sketching provides all columns and rows of the LoRA modules with uniform update frequency in expectation. The overall procedure of FSLoRA is summarized in Algorithm 1.

**Remark 3.5** (Aggregation). Existing works on federated LoRA primarily adopt two aggregation strategies: (1) aggregating the LoRA modules as $[\mathbf{B}; \mathbf{A}]$ (e.g., vanilla Federated LoRA Zhang et al. (2024)), and (2) aggregating the product $\mathbf{BA}$ (e.g., FlexLoRA Bai et al. (2024)). Both methods have demonstrated effectiveness, as evidenced by their promising performance in prior studies. In this work, we adopt the former, as it introduces minimal overhead and retains the simplicity of LoRA. Additionally, we establish the convergence of FSLoRA under this aggregation choice in Section 4. We also demonstrate that FSLoRA is compatible with secure aggregation in Appendix D.

**Remark 3.6** (Computation, memory, and communication). The proposed FSLoRA introduces no additional operations compared to the vanilla Federated LoRA (Zhang et al., 2024), resulting in minimal overhead for both the server and the clients relative to other heterogeneous LoRA baselines (Bai et al., 2024; Wang et al., 2024). A more detailed comparison is provided in Appendix A.

### 3.4 Comparison with Communication Compression

Although both the sketching approach in FSLoRA and communication compression (Kuo et al., 2024) reduce communication overhead, the sketching approach fundamentally differs from traditional compression techniques. Compression methods focus solely on reducing the transmission load, leaving the gradient computation and model updates unchanged from the vanilla Federated LoRA. FSLoRA goes beyond communication savings by also reducing gradient computation and model update overhead through sparse training. Notably, these two methods are orthogonal and can be combined to achieve greater efficiency. Specifically, the compression can be applied to the transmission of non-zero columns of $\mathbf{B}$ and the non-zero rows of $\mathbf{A}$ in FSLoRA to further enhance communication efficiency. We demonstrate the effectiveness of this combination in Appendix H.4.

## 4 Analysis

In this section, we analyze the convergence of the proposed FSLoRA algorithm. We show that the iterate sequence generated by the FSLoRA algorithm converges to a stationary point of the function (2). Our analysis relies on the following notations.

**Notations:** We define $\tilde{\ell}(\mathbf{B}, \mathbf{A}, \xi; \mathbf{S}) = \ell(\mathbf{W}_0 + \mathbf{BSA}, \xi)$ and $\tilde{f}_i(\mathbf{B}, \mathbf{A}; \mathbf{S}) = \mathbb{E}_{\xi \sim \mathcal{D}_i}[\ell(\mathbf{W}_0 + \mathbf{BSA}, \xi)]$ for a given $\mathbf{S}$ and $f_i^{\mathcal{S}}(\mathbf{B}, \mathbf{A}) = \mathbb{E}_{\mathbf{S} \sim \mathcal{S}_i}[\tilde{f}_i(\mathbf{B}, \mathbf{A}; \mathbf{S})]$. For simplic-

ity, we denote $\mathbf{X} = [\mathbf{B}; \mathbf{A}]$ and rewrite $f(\mathbf{B}, \mathbf{A})$, $f_i(\mathbf{B}, \mathbf{A})$, $f^{\mathcal{S}}(\mathbf{B}, \mathbf{A})$, $f_i^{\mathcal{S}}(\mathbf{B}, \mathbf{A})$, $\tilde{f}_i(\mathbf{B}, \mathbf{A}; \mathbf{S})$, and $\tilde{\ell}(\mathbf{B}, \mathbf{A}, \xi; \mathbf{S})$ as $f(\mathbf{X})$, $f_i(\mathbf{X})$, $f^{\mathcal{S}}(\mathbf{X})$, $f_i^{\mathcal{S}}(\mathbf{X})$, $\tilde{f}_i(\mathbf{X}; \mathbf{S})$, and $\tilde{\ell}(\mathbf{X}, \xi; \mathbf{S})$ respectively. In addition, we use $\|\cdot\|$ to denote the Frobenius norm.

We conduct analysis based on the following assumptions.

**Assumption 4.1.** $f_i(\mathbf{X})$ is differentiable and $L$-smooth, i.e., there exists a positive constant $L$ such that $\forall \mathbf{X}, \mathbf{Y}$,

$$\|\nabla f_i(\mathbf{X}) - \nabla f_i(\mathbf{Y})\| \leq L\|\mathbf{X} - \mathbf{Y}\|, \forall i. \tag{8}$$

**Assumption 4.2.** $\nabla_{\mathbf{X}}\tilde{\ell}(\mathbf{X}, \xi; \mathbf{S})$ is an unbiased estimate of $\nabla_{\mathbf{X}}f_i^{\mathcal{S}}(\mathbf{X})$ and its variance is bounded as

$$\mathbb{E}\|\nabla_{\mathbf{X}}\tilde{\ell}(\mathbf{X}, \xi; \mathbf{S}) - \nabla_{\mathbf{X}}f_i^{\mathcal{S}}(\mathbf{X})\|^2 \leq \rho\|\nabla_{\mathbf{X}}f_i^{\mathcal{S}}(\mathbf{X})\|^2 + \sigma^2, \forall i, \tag{9}$$

where the expectation is computed over $\xi \sim \mathcal{D}_i$ and $\mathbf{S} \sim \mathcal{S}_i$.

**Assumption 4.3.** The gradient dissimilarity between the global loss $f^{\mathcal{S}}(\mathbf{X})$ and each local loss $f_i^{\mathcal{S}}(\mathbf{X})$ satisfies

$$\left\|\nabla_{\mathbf{X}}f_i^{\mathcal{S}}(\mathbf{X}) - \nabla_{\mathbf{X}}f^{\mathcal{S}}(\mathbf{X})\right\|^2 \leq c_h\|\nabla_{\mathbf{X}}f^{\mathcal{S}}(\mathbf{X})\|^2 + \delta_h^2, \forall i, \tag{10}$$

where $c_h \geq 0$ and $f^{\mathcal{S}}(\mathbf{X}) = \frac{1}{N}\sum_{i=1}^{N}f_i^{\mathcal{S}}(\mathbf{X})$.

Assumption 4.1 is standard in optimization literature (Bottou et al., 2018; Fang et al., 2024; Bubeck et al., 2015). Assumptions 4.2 and 4.3 are commonly adopted in federated learning to bound sampling randomness and data heterogeneity (Fang et al., 2022; Yi et al., 2022). We further provide an empirical validation in Appendix E, showing that Assumptions 4.2 and 4.3 are reasonable within the LLM fine-tuning scenario. Building on these assumptions, we analyze the convergence behavior of FSLoRA. Our main results are summarized in the following theorem.

**Theorem 4.4.** *Suppose that Assumptions 4.1-4.3 hold, then there exists a learning rate $\gamma \leq \min\{\frac{N}{24\rho(c_h+1)H\bar{L}}, \frac{1}{8\sqrt{\widetilde{L}L(\rho+1)(c_h+1)}H}, \frac{1}{H}\}$ such that the iterates $\{\mathbf{X}^t\}$ generated by FSLoRA satisfy*

$$\frac{1}{T}\sum_{t=0}^{T-1}\mathbb{E}\left\|\nabla_{\mathbf{X}}f^{\mathcal{S}}(\mathbf{X}^t)\right\|^2 \leq 8\frac{\sqrt{\bar{L}\mathcal{F}_0\sigma_\rho^2}}{\sqrt{NTH}} + 10(\widetilde{L}L)^{\frac{1}{3}}\left(\frac{\mathcal{F}_0\sigma_\rho}{T}\right)^{\frac{2}{3}} + \frac{4\mathcal{F}_0}{T}, \tag{11}$$

*where $\sigma_\rho^2 = \sigma^2 + 3(\rho+1)\delta_h^2$, $\bar{L} = \left(\frac{1}{N}\sum_{i=1}^{N}\frac{r}{k_i}\right)L$, $\widetilde{L} = \left(\frac{1}{N}\sum_{i=1}^{N}\frac{r^2}{k_i^2}\right)L$ and $\mathcal{F}_0 = f^{\mathcal{S}}(\mathbf{X}^0) - f^*$ with $f^*$ denoting the lower bound of $f^{\mathcal{S}}(\mathbf{X})$.*

**Technical highlights of Theorem 4.4:** A key step in the proof of Theorem 4.4 is characterizing the impact of the sketching mechanism on the optimization landscape. Our analysis reveals how the sketching operation modifies the smoothness properties of the objective, introducing scaled smoothness constants, $\frac{r}{k_i}L$ and $\frac{r^2}{k_i^2}L$, which directly influence the convergence behavior. Further details are presented in Appendix J.3.

**Discussion:** Theorem 4.4 establishes an upper bound on the convergence of the proposed FSLoRA algorithm. The parameters $\bar{L}$ and $\widetilde{L}$ provide insight into how the sketching operation influences the convergence rate. Increasing $k_i$ would lead to a faster convergence for FSLoRA. However, this comes at the cost of increased communication and computational overhead for client $i$, indicating a trade-off in the selection of the sketching ratios. Additionally, the upper bound vanishes as $T \to \infty$. Moreover, the rate at which the bound diminishes is dominated by the first term, which recovers the convergence behavior of FedAvg (Yu et al., 2019; Khaled et al., 2020; Karimireddy et al., 2020) as the sketching ratio $k_i/r \to 1$(i.e., $\bar{L} = L$). This highlights the tightness of our analysis and shows that FSLoRA retains the convergence guarantees of vanilla Federated LoRA in the limit.

## 5 EXPERIMENTS

Our experiments focus on RoBERTa (125M) (Liu, 2019) and LLaMA-3.2-3B (Dubey et al., 2024), which represent typical model sizes suitable for client-side deployment, as well as the LLaMA-7B model to reflect large-scale scenarios. For RoBERTa and LLaMA-3.2-3B models, we fine-tune and evaluate them on the GLUE (Wang, 2018) and commonsense reasoning benchmark (Hu et al.,

Table 5.1: Testing accuracy over 3 independent runs on GLUE and commonsense reasoning benchmarks. FSLoRA achieves a notable improvement in average performance compared to the baselines.

| GLUE benchmark (RoBERTa model) | | | | | | | | | |
|---|---|---|---|---|---|---|---|---|---|
| Method | GPU hours | QNLI | MRPC | CoLA | MNLI | RTE | SST-2 | QQP | Avg. |
| HeteroLoRA | 10.7h | 87.5 ±0.5 | 84.4 ±0.9 | 75.3 ±1.2 | 66.3 ±0.8 | 69.0 ±1.7 | 89.5 ±0.0 | 85.3 ±0.1 | 79.6 |
| FlexLoRA | 12.6h | 88.5 ±0.2 | 81.2 ±0.4 | 77.5 ±1.2 | 63.0 ±0.5 | 62.2 ±1.9 | 92.8 ±0.4 | 87.4 ±0.1 | 78.9 |
| FLoRA | 12.3h | 87.2 ±0.3 | 78.1 ±0.7 | 77.4 ±1.7 | 74.6 ±0.5 | 54.4 ±2.1 | 93.4 ±0.1 | 87.1 ±0.3 | 78.9 |
| FSLoRA | 10.9h | 88.0 ±0.3 | 87.3 ±0.2 | 82.2 ±0.5 | 76.4 ±0.2 | 69.8 ±1.2 | 93.5 ±0.1 | 85.8 ±0.1 | 83.3 |

| Commonsense reasoning benchmark (LLaMA-3.2-3B model) | | | | | | | | | |
|---|---|---|---|---|---|---|---|---|---|
| Method | GPU hours | ARC-c | ARC-e | BoolQ | HellaSwag | OBQA | PIQA | SIQA | WinoGrande | Avg. |
| HeteroLoRA | 43.7h | 73.4 ±0.3 | 86.6 ±0.2 | 65.8 ±0.5 | 73.0 ±0.5 | 71.4 ±0.3 | 80.9 ±0.7 | 73.8 ±0.3 | 72.0 ±0.3 | 74.6 |
| FlexLoRA | 68.3h | 74.2 ±0.3 | 86.7 ±0.6 | 68.6 ±0.8 | 79.4 ±0.7 | 75.8 ±0.4 | 81.0 ±0.3 | 75.9 ±0.4 | 77.9 ±0.3 | 77.4 |
| FLoRA | 49.8h | 68.3 ±0.6 | 83.1 ±0.5 | 65.8 ±0.9 | 77.2 ±0.5 | 74.2 ±0.3 | 80.5 ±0.6 | 76.1 ±0.5 | 71.5 ±0.5 | 74.6 |
| FSLoRA | 44.3h | 76.1 ±0.4 | 87.2 ±0.5 | 69.3 ±0.7 | 82.2 ±1.1 | 80.7 ±0.6 | 84.0 ±0.2 | 76.8 ±0.0 | 79.1 ±0.2 | 79.4 |

2023), respectively. For the LLaMA-7B model, we utilize Wizard, Dolly-15k, and Alpaca datasets, where the results are reported in Appendix G. Similar to (Zhang et al., 2024; Wang et al., 2024), we adopt Dirichlet-based partitioning for dataset splits. All the experiments are conducted on a cluster equipped with 4 NVIDIA A100 GPUs, each with 40 GB of memory. The number of clients is set to 20 in the main manuscript, and to 50 and 100 in Appendix F. Further details are provided in the Appendix I. The implementation code for this project is included in the supplementary material.

## 5.1 MAIN RESULTS UNDER HETEROGENEOUS LORA SETUP

**Performance comparison with baselines:** We consider three state-of-the-art baselines listed in Section 2.2. For FSLoRA, the rank of the global LoRA modules is fixed as $r = 64$, while the sketching ratio for client $i$ is sampled from the set $\{0.125, 0.25, 0.5, 0.75\}$. For a fair comparison, we apply the same rank configuration to all baseline methods. Table 5.1 presents the performance of FSLoRA and baseline methods. Across both settings, FSLoRA consistently achieves superior accuracy while maintaining low GPU hours. In the GLUE & RoBERTa task, FSLoRA outperforms all baselines on average, with significant gains in MRPC, CoLA, and MNLI. In the commonsense reasoning & LLaMA task, which introduces higher model complexity, FSLoRA also delivers the best overall performance. Notably, FSLoRA achieves this while preserving computational efficiency comparable to HeteroLoRA as reflected in GPU hours. These results highlight FSLoRA's effectiveness and scalability in heterogeneous LoRA fine-tuning scenarios.

**Evaluation under broader heterogeneity, increased number of clients, and larger model:** In Appendix F, we extend our evaluation to 50 and 100 clients, incorporating greater diversity in clients' communication and computation capabilities, as well as varying levels of data heterogeneity. In Appendix G, we further assess the effectiveness of our method on the LLaMA-7B model.

## 5.2 ABLATION STUDY

**Impact of sketching:** In Figures 2 and 3(a), we compare the performance of FSLoRA with and without sketching on fine-tuning the RoBERTa model and the LLaMA-3.2-3B model, respectively. Notably, FSLoRA without sketching is equivalent to the vanilla Federated LoRA. For FSLoRA with sketching, we apply a uniform sketching ratio of $k_i/r = 0.5$ across all distributed clients. The upload budget for each client is set to 100 and 80 times the size of the full global LoRA modules at the corresponding rank for the RoBERTa and the LLaMA-3.2-3B models, respectively. As shown in Figures 2 and 3(a), both FSLoRA with and without sketching achieve higher accuracy when the rank $r$ increases due to the availability of more tunable parameters. In addition, FSLoRA consistently outperforms its non-sketched counterpart across all the ranks and datasets. The use of sketching increases the communication frequency for clients under the same communication budget, thereby facilitating the optimization process and enhancing fine-tuning efficiency.

**Impact of the global rank:** In Figure 3(b), we investigate the impact of the rank of the global LoRA modules on FSLoRA's performance. We vary the rank of the global LoRA modules while keeping the rank of submatrices updated by the clients to be consistent (i.e., $k_i = 8$). This ensures that the communication and computational resources on the client side remain unchanged. As illustrated in Figure 3(b), FSLoRA maintains stable convergence across all the configurations. Moreover, FSLoRA

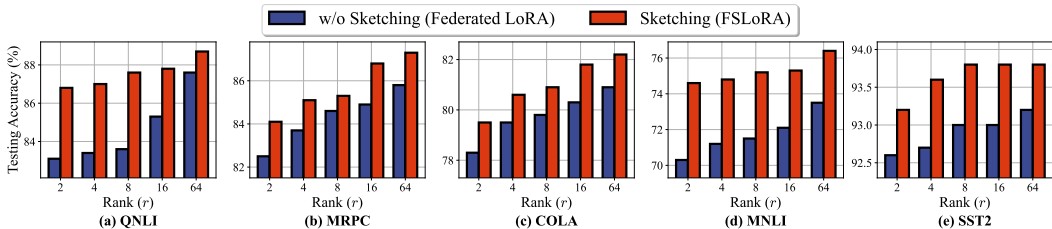

Figure 2: Comparison between FSLoRA with and without sketching (the latter equivalent to Federated LoRA) where the upload budget for clients is set to $100\times$ the size of the global LoRA modules at each rank. FSLoRA obtains a better performance, validating its communication efficiency.

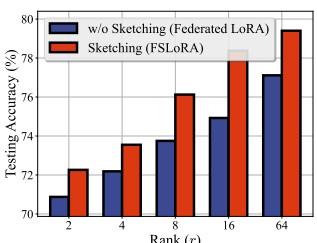 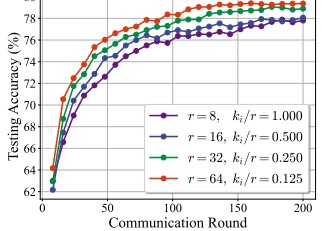 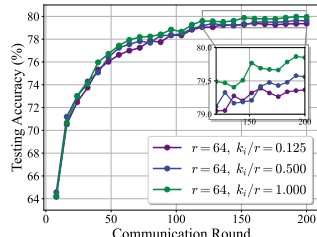

(a) Comparison of FSLoRA with and without sketching, with an upload budget $80\times$ the size of the global LoRA modules at each rank.

(b) Impact of the rank of global LoRA modules on FSLoRA, given a fixed rank $k_i$ for the updated submatrices at the clients.

(c) Impact of the sketching ratio on FSLoRA's performance under a fixed rank $r = 64$ for the global LoRA modules.

Figure 3: Fine-tuning the LLaMA-3.2-3B model on the commonsense reasoning benchmark. The results are averaged over eight tasks, illustrating FSLoRA's ability to maintain strong performance while adapting to different rank and sketching configurations.

demonstrates improved performance as the global rank increases. This observation confirms that the proposed sketching mechanism enables resource-constrained systems to reap the benefits of a higher global rank, striking an effective balance between efficiency and performance.

**Impact of sketching ratio:** Finally, we investigate the impact of the sketching ratio on FSLoRA's performance by maintaining a constant global LoRA rank $r = 64$ while varying the sketching ratio $k_i/r$ in the range $\{0.125, 0.5, 1\}$. As shown in Figure 3(c), there is a slight performance degradation as the sketching ratio decreases, which is consistent with our theoretical analysis. This reflects an inherent tradeoff: while a larger sketching ratio improves convergence and accuracy, a smaller ratio reduces both computational and communication overhead. Notably, the observed degradation remains minor, highlighting FSLoRA's ability to maintain strong performance even under constrained resources. This demonstrates its effectiveness in balancing efficiency and accuracy, making it well-suited for resource-limited scenarios.

**Further experiments:** Results with more clients under broader heterogeneity, as well as with a larger model, are reported in Appendix F and Appendix G, respectively. Appendix H.1 provides detailed per-task comparisons on the commonsense reasoning benchmark corresponding to Figures 3(a) and 3(b). The impact of varying the number of local updates $H$ is studied in Appendix H.2, while the extension to dynamic sketching ratios is presented in Appendix H.3. Finally, Appendix H.4 demonstrates the synergistic effect of integrating compression with sketching.

## 6 CONCLUSION

We have proposed FSLoRA, a novel collaborative LLM fine-tuning framework that introduces a sketching mechanism to enhance both performance and efficiency in resource-constrained systems. By maintaining large-rank LoRA modules on the server and allowing clients to selectively update submatrices based on the sketching ratios, FSLoRA effectively adapts to heterogeneous communication and computational constraints. We provide a rigorous convergence analysis of FSLoRA that characterizes how the sketching ratios affect the convergence rate. Finally, we confirmed the effectiveness of FSLoRA through extensive experiments across multiple datasets and models.

## LIMITATION

A potential limitation is that our paper is primarily theoretical. Extending the proposed techniques to practical network environments and evaluating their behavior under unstable connections, delayed clients, and latency fluctuations remains an important direction for future work.

## REPRODUCIBILITY STATEMENT

This paper provides all the necessary information to reproduce the main experimental results. The datasets used are publicly available, while the model architectures, hyperparameters, and training details are documented in Section 5. The full implementation code is included in the supplementary material of our submission.

## LLM USAGE

We used ChatGPT (GPT-5), as an assistive tool only for improving the clarity and readability of the manuscript. The LLM was employed to polish grammar and rephrase sentences for conciseness. It was not used for research ideation, methodological design, and experimental implementation. All scientific content, including problem formulation, algorithm development, analysis, and experiments, was entirely conceived and executed by the authors.

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

# Appendix

# A COMPARISON OF COMPUTATION, MEMORY, AND COMMUNICATION

**Computation and memory:** Let $P$ and $q$ denote the memory cost of the full model and the global LoRA module (rank $r$), respectively. The computational cost is expressed with the big O notation. Forward and backward computations, as well as activation memory, are omitted as they are identical across all the considered methods. The results are summarized in Tables A.1 and A.2, where $m$ and $n$ denote the shape of the base model, $k_i$ denotes the LoRA rank for client $i$, $H$ denotes the number of iterations per round, and $N$ is the number of clients. Additionally, the results for the vanilla Federated LoRA, denoted as FedLoRA, are reported under the case of homogeneous LoRA ranks, i,e., $k_i = r$.

Table A.1: Client-side computation load and memory usage comparison.

| Method | Memory | Computation (per round) |
|---|---|---|
| FedLoRA | $P + q$ | $\mathcal{O}(Hr(m+n))$ |
| HeteroLoRA | $P + \frac{k_i}{r}q$ | $\mathcal{O}(Hk_i(m+n))$ |
| FlexLoRA | $P + \frac{k_i}{r}q$ | $\mathcal{O}(Hk_i(m+n))$ |
| FLoRA | $P + \max\left\{\sum_{i=1}^{N}\frac{k_i}{r}q, P\right\}$ | $\mathcal{O}\left(Hk_i(m+n)) + (\sum_{i=1}^{N}k_i)mn + mn\right)$ |
| FSLoRA | $P + \frac{k_i}{r}q$ | $\mathcal{O}(Hk_i(m+n))$ |

Table A.2: Server-side computation load and memory usage comparison.

| Method | Memory | Computation (per round) |
|---|---|---|
| FedLoRA | $Nq$ | $\mathcal{O}(N(m+n)r)$ |
| HeteroLoRA | $\sum_{i=1}^{N}\frac{k_i}{r}q$ | $\mathcal{O}(N(m+n)r)$ |
| FlexLoRA | $\max\left\{\sum_{i=1}^{N}\frac{k_i}{r}q, 2P\right\}$ | $\mathcal{O}\left((\sum_{i=1}^{N}k_i)mn + Nmn + \min\{m,n\}mn\right)$ |
| FLoRA | $\sum_{i=1}^{N}\frac{k_i}{r}q$ | $\mathcal{O}\left((\sum_{i=1}^{N}k_i)(m+n)\right)$ |
| FSLoRA | $\sum_{i=1}^{N}\frac{k_i}{r}q$ | $\mathcal{O}(N(m+n)r)$ |

As shown in Tables A.1 and A.2, FSLoRA matches HetLoRA in both computation and memory cost. FLoRA introduces additional client-side overhead due to merging LoRA modules. FlexLoRA incurs extra server-side costs from conducting SVD on the full model. In summary, FSLoRA guarantees convergence with minimum overhead.

**Communication:** We detailed the communication load for baselines and our methods in Table A.3, where $q$ denotes the communication cost of a global LoRA module with rank $r$, $k_i$ denotes the local LoRA rank for client $i$, $m$ and $n$ denote the shape of the base model, and $N$ denotes the number of clients.

Table A.3: Communication complexity, assuming float 32 parameters and binary sketching indices.

| | FedLoRA | HeteroLoRA | FlexLoRA | FLoRA | FSLoRA |
|---|---|---|---|---|---|
| Uplink | $q$ | $\frac{k_i}{r}q$ | $\frac{k_i}{r}q$ | $\frac{k_i}{r}q$ | $\frac{k_i}{r}q$ |
| Downlink | $q$ | $q$ | $q$ | $\sum_{i=1}^{N}\frac{k_i}{r}q$ | $q(1 + \frac{Nr}{32mn})$ |

For the uplink, all four heterogeneous LoRA algorithms incur the same communication overhead for transmitting updated local LoRA modules, which is lower than that of FedLoRA. For the downlink, FLoRA requires broadcasting the stacked LoRA modules, while HeteroLoRA and FlexLoRA broadcast the updated global LoRA modules. FSLoRA, on the other hand, broadcasts both the global LoRA modules and additional sketching matrices. The extra communication introduced by the sketching matrices is negligible compared to that of the global LoRA modules, as it consists only of *binary sketching indices* (i.e., the diagonal elements of the sketching matrix). For instance, in the case of the LLaMA-3.2-3B model under our experimental LoRA configuration, the global LoRA modules contain 66,060,288 parameters, equivalent to approximately 252 MB when using float32. With a

global rank of $r = 64$, the sketching indices require only 64 bits per client, covering all LoRA layers. Even with 100 clients, the total sketching overhead is merely 0.78 KB, which accounts for only 0.0003% of the global LoRA modules.

## B  DIFFERENCE BETWEEN FSLoRA AND FedBCGD

Both FSLoRA and federated block coordinate gradient descent (FedBCGD) (Liu et al., 2024) are motivated by client heterogeneity but are designed for fundamentally different deployment contexts. FedBCGD partitions the full model $\mathbf{x} = [\mathbf{x}_1, \ldots, \mathbf{x}_N, \mathbf{x}_s]$, assigning each block $\mathbf{x}_j$ to a subset of clients with similar resource constraints, while the shared block $\mathbf{x}_s$ is optimized across all clients. While this block-partitioning strategy is effective for smaller models, it relies on explicit and static allocation, which can limit scalability and flexibility. As such, FedBCGD and similar block coordinate methods based on the full model are less suitable for federated LLM fine-tuning.

FSLoRA, in contrast, builds on LoRA and introduces sparse diagonal sketching. Given a sketch matrix $\mathbf{S}$, the gradients of the loss $\ell(\mathbf{W}_0 + \mathbf{BSA}, \xi)$ with respect to the LoRA matrices $\mathbf{B}$ and $\mathbf{A}$ are sparse: only selected columns of $\mathbf{B}$ and rows of $\mathbf{A}$ are updated in each round. By configuring the rank and sparsity of the sketch matrix $\mathbf{S}$, FSLoRA flexibly controls both the computational and communication load per client, enabling adaptation to heterogeneous client capabilities.

The distinctions between FedBCGD and FSLoRA are summarized in Table B.1. To wrap up, these two algorithms are tailored for distinct purposes and deployment contexts.

Table B.1: Conceptual distinctions between FSLoRA and FedBCGD.

| Aspect | FedBCGD | FSLoRA |
|---|---|---|
| Partition Type | Explicit & static | Random & sketching-based |
| Model Scope | Full model | LoRA modules |
| Adaptation Strategy | Assign different blocks | Adjust sketch rank (sparsity) |

## C  JUSTIFICATION FOR RANDOM-$k$ SKETCHING

FSLoRA is built upon Random-$k$ diagonal sketching due to two key properties:

- **Submatrix selection.** Given a sparse diagonal matrix $\mathbf{S}_i$, we have

$$\mathbf{BS}_i\mathbf{A} = \sum_{j \in \mathcal{I}_i} s_j \mathbf{b}_j \mathbf{a}_j^\top = [\mathbf{b}_j]_{j \in \mathcal{I}_i} \operatorname{diag}\{s_j\}_{j \in \mathcal{I}_i} [\mathbf{a}_j^\top]_{j \in \mathcal{I}_i}, \qquad (12)$$

  where $\mathcal{I}_i$ corresponds to the index set of non-zero diagonal entries of $\mathbf{S}_i$ and $\mathbf{b}_j$ and $\mathbf{a}_j^\top$ denote the $j$-th column of module $\mathbf{B}$ and the $j$-th row of module $\mathbf{A}$, respectively. In other words, with Random-$k$ sketching, only a subset of $\mathbf{B}$'s columns and $\mathbf{A}$'s rows are activated for client $i$. The *sparse diagonal structure* effectively reduces local training cost for each client.

- **Unbiasedness for convergence.** When $\mathbf{S}_i$ is a Random-$k$ diagonal matrix with $k_i$ nonzero diagonal entries $s_j = \frac{r}{k_i}$, we have
$$\mathbb{E}[\mathbf{BS}_i\mathbf{A}] = \mathbf{BA}. \qquad (13)$$
  This unbiasedness is critical for our convergence analysis.

Table C.1: Accuracy comparison of Random-$k$ sketching and importance-based sketching under the commonsense reasoning benchmark with the LLaMA-3.2-3B model. Random-$k$ sketching achieves better performance.

| Importance metric | ARC-c | ARC-e | BoolQ | HSwag | OBQA | PIQA | SIQA | Wino | Avg. |
|---|---|---|---|---|---|---|---|---|---|
| $\|a\|\|b\|$ | 71.9 | 86.5 | 55.2 | 75.4 | 73.4 | 81.1 | 72.5 | 69.7 | 73.2 |
| $\|a\| + \|b\|$ | 72.1 | 86.4 | 64.5 | 76.8 | 70.8 | 82.2 | 71.3 | 69.3 | 74.2 |
| Random-$k$ | 75.8 | 86.7 | 69.7 | 81.4 | 80.4 | 83.9 | 76.2 | 78.8 | 79.1 |

In addition, we experimentally compare Random-$k$ sketching with importance-based sketching. For importance-based sketching, we sample (sketch) $k_i$ components from $\{\mathbf{b}_j \mathbf{a}_j^\top\}_{j=1}^r$ with probability set as the importance scores, e.g., $\|\mathbf{b}_j\|_2 + \|\mathbf{a}_j\|_2$ or spectral norm $\|\mathbf{b}_j\|_2 \cdot \|\mathbf{a}_j\|_2$. The results are shown in Table C.1. The results show that Random-$k$ diagonal sketching outperforms these choices for importance-based sketching. This may be because these importance measures are heuristic and do not reliably reflect actual contribution. Moreover, such sketching violates the unbiasedness property, complicating theoretical guarantees. While improved importance-based methods may enhance performance and remain a promising direction for future investigation, our current empirical and theoretical results favor random sketching.

## D  COMPATIBILITY OF FSLoRA WITH SECURE AGGREGATION

The aggregation of FSLoRA is compatible with secure aggregation. Taking the aggregation of module $\mathbf{B}$ as an example, we illustrate this below.

In FSLoRA, client updates are sparse matrices with non-zero values only in columns indexed by $\mathcal{I}_i \subset [r]$, size $|\mathcal{I}_i| = k_i$. With secure aggregation, each client apply *additive masking*:

$$\tilde{\mathbf{B}}_i = \Delta \mathbf{B}_i + \mathbf{R}_i, \tag{14}$$

where mask $\mathbf{R}_i$ satisfies $\mathrm{supp}(\mathbf{R}_i) \subseteq (u,v) : u \in [m], v \in \mathcal{I}_i$ and $\sum_{i=1}^N \mathbf{R}_i = 0$. That is, the mask has non-zero entries only in the client's active columns, and all masks together sum to zero to preserve correctness. Such masks can be constructed following the classical protocol: for each pair of clients $(i,j)$, define a random matrix

$$\mathbf{M}_{ij} = -\mathbf{M}_{ji} \in \mathbb{R}^{m \times r}, \quad \mathrm{supp}(\mathbf{M}_{ij}) \subseteq (u,v) \mid u \in [m],\, v \in \mathcal{I}_i \cap \mathcal{I}_j, \tag{15}$$

and then construct its total mask

$$\mathbf{R}_i = \sum_{j>i} \mathbf{M}_{ij} - \sum_{j<i} \mathbf{M}_{ji}. \tag{16}$$

During uploading, client $i$ sends the masked $k_i$ non-zero columns of $\tilde{\mathbf{B}}_i$, and then the server adds the corresponding padding and averages them as:

$$\sum_{i=1}^N \tilde{\mathbf{B}}_i = \sum_{i=1}^N (\Delta \mathbf{B}_i + \mathbf{R}_i) = \sum_{i=1}^N \Delta \mathbf{B}_i, \tag{17}$$

which matches the aggregation of module $\mathbf{B}$ in (7). The aggregation of module $\mathbf{B}$ in FSLoRA is thus compatible with secure aggregation.

We can draw the same conclusion for module $\mathbf{A}$ under the same derivation.

## E  EMPIRICAL VALIDATION OF ASSUMPTIONS

In the context of LLM fine-tuning, both the magnitude of stochastic gradients and gradients in LLM fine-tuning are in a mild range since:

- Transformer-based LLMs use *LayerNorm* and *scaled softmax attention*, which stabilize activations and suppress gradient spikes.
- Fine-tuning starts from a well-pretrained model already near a local minimum, leading to smaller gradients.
- The fine-tuning dataset does not typically contain strong contradictory signals to what the model already knows, resulting in a relatively flat loss surface.

To further support this empirically, we report the statistics of the expected norm of the stochastic gradients $\mathbb{E}\|\nabla_{\mathbf{X}} \tilde{\ell}(\mathbf{X}, \xi; \mathbf{S})\|$ over approximately $4500$ samples for $4$ representative clients with different sketching ratios. The table below reports the minimum and maximum expected norm among $30$ randomly sampled model states $\mathbf{X} = [\mathbf{B}; \mathbf{A}]$.

Table E.1: Statistics of the expected norm of stochastic gradients across clients.

| Client ID | Number of samples | Rank | Min | Max |
|:---:|:---:|:---:|:---:|:---:|
| 1 | 4580 | 4 | 0.1286 | 0.9499 |
| 2 | 4216 | 19 | 0.1284 | 0.7069 |
| 3 | 4873 | 9 | 0.1237 | 0.5774 |
| 4 | 5124 | 32 | 0.1499 | 0.8066 |

As we can see from Table E.1, the expected norm, i.e., $\mathbb{E}_{\xi \sim \mathcal{D}_i, \mathbf{S} \sim \mathcal{S}_i} \|\nabla_{\mathbf{X}} \tilde{\ell}(\mathbf{X}, \xi; \mathbf{S})\|$, is in a moderate range. Notably, the variance is upper-bounded by the expected squared gradient norm:

$$\mathbb{E}_{\xi \sim \mathcal{D}_i, \mathbf{S} \sim \mathcal{S}_i} \|\nabla_{\mathbf{X}} \tilde{\ell}(\mathbf{X}, \xi; \mathbf{S}) - \nabla_{\mathbf{X}} f_i^{\mathcal{S}}(\mathbf{X})\|^2 \leq \mathbb{E}_{\xi \sim \mathcal{D}_i, \mathbf{S} \sim \mathcal{S}_i} \|\nabla_{\mathbf{X}} \tilde{\ell}(\mathbf{X}, \xi; \mathbf{S})\|^2.$$

Therefore, it is generally not hard to find a $\sigma$ and $\rho$ to make Assumption 4.2 hold.

On the other hand, we have

$$
\begin{aligned}
& \left\|\nabla_{\mathbf{X}} f_i^{\mathcal{S}}(\mathbf{X}) - \nabla_{\mathbf{X}} f^{\mathcal{S}}(\mathbf{X})\right\|^2 \\
\leq & 2\left\|\nabla_{\mathbf{X}} f_i^{\mathcal{S}}(\mathbf{X})\right\|^2 + 2\left\|\nabla_{\mathbf{X}} f^{\mathcal{S}}(\mathbf{X})\right\|^2 \\
\leq & 2\left\|\nabla_{\mathbf{X}} f_i^{\mathcal{S}}(\mathbf{X})\right\|^2 + 2\frac{1}{N}\sum_{i=1}^{N}\left\|\nabla_{\mathbf{X}} f_i^{\mathcal{S}}(\mathbf{X})\right\|^2 \\
\leq & 2\mathbb{E}_{\xi \sim \mathcal{D}_i, \mathbf{S} \sim \mathcal{S}_i}\|\nabla_{\mathbf{X}} \tilde{\ell}(\mathbf{X}, \xi; \mathbf{S})\|^2 + 2\frac{1}{N}\sum_{i=1}^{N}\mathbb{E}_{\xi \sim \mathcal{D}_i, \mathbf{S} \sim \mathcal{S}_i}\|\nabla_{\mathbf{X}} \tilde{\ell}(\mathbf{X}, \xi; \mathbf{S})\|^2,
\end{aligned}
\tag{18}
$$

where the first inequality follows Cauchy-Schwarz inequality, while the last inequality follows Jensen's inequality. Thus, the deviation can be controlled by the expected gradient norm, which we empirically found to be moderate (see Table E.1). Hence, it is reasonable to impose an upper bound on $\left\|\nabla_{\mathbf{X}} f_i^{\mathcal{S}}(\mathbf{X}) - \nabla_{\mathbf{X}} f^{\mathcal{S}}(\mathbf{X})\right\|^2$ as in Assumption 4.3.

# F   EVALUATION UNDER BROADER HETEROGENEITY AND INCREASED NUMBER OF CLIENTS

To accommodate a larger number of clients, we extend FSLoRA (Algorithm 1) to support *partial client participation*. Specifically, at each round, the server samples a subset of clients, distributes the current global LoRA modules to them, and aggregates only the updates from these clients. Throughout this section, we fix the partial participation size to 10, i.e., 10 clients are sampled in each round.

## F.1   INCREASING RESOURCE HETEROGENEITY AND THE NUMBER OF CLIENTS

We extend our experiments on LLaMA-3.2-3B with the commonsense reasoning benchmark to 50 clients. We adopt Dirichlet-based partitioning for dataset splits. Specifically, the commonsense reasoning benchmark includes 8 tasks, and we partitioned them based on the Dirichlet distribution to construct task heterogeneity among 50 clients. The Dirichlet concentration parameter is set to $\alpha = 0.1$. We simulate client resource heterogeneity via different LoRA rank distributions (beyond the limited sketching ratio considered in Section 5). More capable clients are assigned higher ranks, reflecting varying compute capacities. We consider two different rank distributions: normal and heavy-tail distributions in the range $[4, 64]$.

**Normal distribution:** Ranks are sampled from a normal distribution with mean $\mu = \frac{a+b}{2}$ and standard deviation $\sigma = \frac{b-a}{6}$, where $a = 4$ and $b = 64$. This models a balanced distribution of client capabilities centered around the middle of the range.

**Heavy-tail distribution:** We sample ranks using an inverse log-normal distribution. Specifically, we draw $x_i \sim \text{LogNormal}(\mu, \sigma)$ with $\mu = \log\left(\frac{a+b}{4}\right)$ and $\sigma = 1.0$, then set $k_i = 1/x_i$ and apply min-max normalization to scale values into the range $[a, b]$. This results in a heavy-tailed distribution

where most clients receive low ranks, reflecting a scenario with many low-capability clients and a few high-capability ones.

Table F.1: Accuracy comparison under different client heterogeneity settings. FSLoRA outperforms baseline methods across both normal and heavy-tail LoRA rank distributions.

| Rank setup | Method | ARC-c | ARC-e | BoolQ | HellaSwag | OBQA | PIQA | SIQA | WinoGrande | Avg. |
|---|---|---|---|---|---|---|---|---|---|---|
| Normal | HeteroLoRA | 73.38 | 85.82 | 62.17 | 71.23 | 77.40 | 80.14 | 74.72 | 72.53 | 74.67 |
| | FlexLoRA | 74.23 | 87.84 | 68.37 | 79.77 | 76.00 | 82.97 | 75.90 | 78.13 | 77.90 |
| | FLoRA | 68.17 | 83.75 | 64.93 | 75.67 | 71.40 | 77.20 | 71.24 | 70.09 | 72.81 |
| | FSLoRA | 75.77 | 86.95 | 69.67 | 81.53 | 80.60 | 84.06 | 76.20 | 78.85 | 79.20 |
| Heavy-tail | HeteroLoRA | 72.44 | 86.78 | 63.60 | 73.10 | 72.00 | 81.34 | 71.65 | 68.75 | 73.71 |
| | FlexLoRA | 73.04 | 86.70 | 62.23 | 75.57 | 78.00 | 81.12 | 74.77 | 73.32 | 75.59 |
| | FLoRA | 67.92 | 81.90 | 64.90 | 72.77 | 74.00 | 80.41 | 75.28 | 70.24 | 73.43 |
| | FSLoRA | 75.77 | 86.70 | 69.67 | 81.40 | 80.40 | 83.90 | 76.15 | 78.77 | 79.10 |

As shown in Table F.1, FSLoRA outperforms other methods under both heterogeneity settings. As we move from normal to heavy-tail, where more clients are low-resource, overall performance decreases for all methods. However, FSLoRA exhibits the smallest drop, demonstrating stronger robustness to extreme client heterogeneity.

In Figure 4, we compare the convergence behavior of FSLoRA and three baseline methods under the aforementioned two types of client heterogeneity. Under the normal distribution, FlexLoRA exhibits fast initial progress but falls behind FSLoRA in final accuracy, likely due to approximation errors introduced by truncated SVD. This issue is exacerbated in the heavy-tail distribution, where low-rank clients dominate and SVD truncation causes greater distortion, further degrading FlexLoRA's performance. Similarly, HeteroLoRA's reliance on zero-padding reduces optimization efficiency, preventing it from achieving higher accuracy. FLoRA fails to show steady improvement as communication progresses. One potential reason is that frequent model merging and random reinitialization of LoRA modules in each round disrupt the convergence continuity. In contrast, FSLoRA demonstrates robust and stable convergence across both scenarios, achieving the highest overall accuracy.

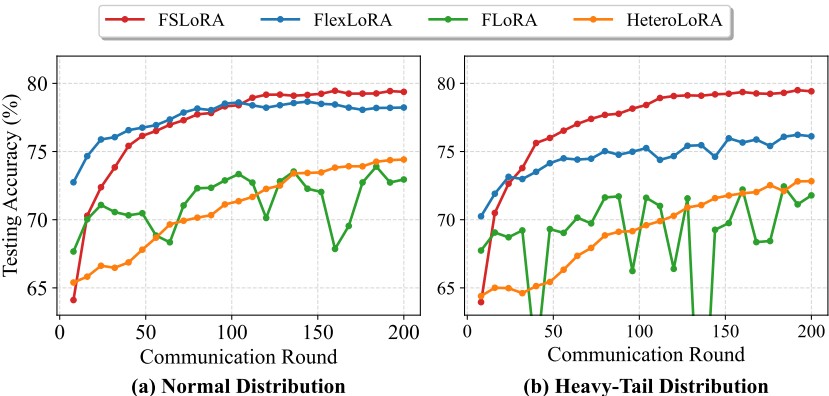

(a) Normal Distribution      (b) Heavy-Tail Distribution

Figure 4: Convergence behavior of FSLoRA and baselines on the commonsense reasoning benchmark with the LLaMA-3.2-3B model. Notably, FSLoRA's per-round communication cost is at most equal to the baselines (as detailed in Appendix A). Testing accuracy is averaged over eight tasks.

## F.2 FURTHER INCREASING THE NUMBER OF CLIENTS

We further evaluated the performance of FSLoRA by increasing the number of clients to 100. The results are presented in Table F.2. In this setting, local ranks follow a heavy-tailed distribution as described in the previous subsection, and all other experimental configurations remain unchanged. As shown in the table, FSLoRA maintains its advantage in terms of the average performance when scaling to more clients.

Table F.2: Accuracy comparison when the number of clients is $N = 100$. FSLoRA maintains its advantage in terms of the average accuracy.

| Method | ARC-c | ARC-e | BoolQ | HellaSwag | OBQA | PIQA | SIQA | WinoGrande | Avg. |
|---|---|---|---|---|---|---|---|---|---|
| HeteroLoRA | 71.76 | 86.24 | 62.57 | 68.07 | 76.60 | 79.38 | 74.10 | 69.69 | 73.55 |
| FlexLoRA | 73.38 | 87.54 | 69.03 | 75.27 | 78.60 | 80.47 | 74.16 | 73.80 | 76.53 |
| FLoRA | 69.97 | 83.25 | 67.10 | 71.67 | 73.60 | 78.94 | 72.21 | 70.80 | 73.44 |
| FSLoRA | 74.40 | 87.54 | 70.13 | 79.90 | 79.40 | 83.57 | 76.51 | 78.93 | 78.80 |

## F.3 VARYING THE LEVEL OF DATA HETEROGENEITY

In Table F.3, we investigate the impact of the degree of data heterogeneity on performance. We increase the heterogeneity by decreasing the Dirichlet concentration parameter from $\alpha = 1$ to $\alpha = 0.1$. The local ranks follow the heavy-tail distribution described in the previous subsection, and all other experimental configurations remain consistent with Appendix F.1. As observed from Table F.3, the performance of all methods degrades as heterogeneity increases. FSLoRA consistently achieves higher accuracy.

Table F.3: Accuracy comparison under different data heterogeneity settings. FSLoRA maintains its advantage over the baselines as the data heterogeneity increases. The number of clients is set to 50.

| Data setup | Method | ARC-c | ARC-e | BoolQ | HellaSwag | OBQA | PIQA | SIQA | WinoGrande | Avg. |
|---|---|---|---|---|---|---|---|---|---|---|
| Dir(1) | HeteroLoRA | 72.18 | 86.11 | 62.57 | 73.10 | 77.60 | 79.82 | 74.26 | 69.46 | 74.39 |
|  | FlexLoRA | 74.06 | 87.25 | 65.67 | 74.90 | 78.80 | 81.01 | 74.16 | 74.27 | 76.27 |
|  | FLoRA | 70.14 | 83.29 | 67.27 | 71.60 | 73.60 | 78.73 | 72.16 | 70.96 | 73.47 |
|  | FSLoRA | 75.85 | 87.50 | 70.93 | 81.47 | 81.00 | 82.86 | 76.66 | 78.53 | 79.35 |
| Dir(0.1) | HeteroLoRA | 72.44 | 86.78 | 63.60 | 73.10 | 72.00 | 81.34 | 71.65 | 68.75 | 73.71 |
|  | FlexLoRA | 73.04 | 86.70 | 62.23 | 75.57 | 78.00 | 81.12 | 74.77 | 73.32 | 75.59 |
|  | FLoRA | 67.92 | 81.90 | 64.90 | 72.77 | 74.00 | 80.41 | 75.28 | 70.24 | 73.43 |
|  | FSLoRA | 75.77 | 86.70 | 69.67 | 81.40 | 80.40 | 83.90 | 76.15 | 78.77 | 79.10 |

# G EXPERIMENTS ON QWEN2.5-1.5B-INSTRUCT AND LLAMA-7B

We further extended our evaluation to the Qwen model. Specifically, we fine-tuned Qwen2.5-1.5B-Instruct on the commonsense reasoning benchmark using the same setup as in Table 5.1. As shown in Table G.1, FSLoRA achieves competitive performance compared with the heterogeneous LoRA baselines, suggesting its effective across different model architectures.

Table G.1: Performance comparison with Qwen2.5-1.5B-Instruct. FSLoRA maintains its advantage.

| Method | ARC-c | ARC-e | BoolQ | HellaSwag | OBQA | PIQA | SIQA | WinoGrande | Avg. |
|---|---|---|---|---|---|---|---|---|---|
| HeteroLoRA | 70.36 | 86.53 | 65.50 | 68.55 | 79.01 | 74.93 | 68.25 | 68.04 | 72.65 |
| FlexLoRA | 73.41 | 88.07 | 63.32 | 71.30 | 76.44 | 78.02 | 70.94 | 68.56 | 73.76 |
| FLoRA | 70.28 | 84.51 | 60.90 | 68.82 | 69.58 | 73.45 | 67.53 | 66.27 | 70.17 |
| FSLoRA | 75.82 | 88.16 | 65.08 | 74.93 | 79.82 | 80.72 | 72.62 | 71.29 | 76.06 |

Although our primary focus is on models suitable for client-side deployment, such as RoBERTa and LLaMA-3.2-3B models, we also include experiments on the larger LLaMA-7B model to demonstrate the scalability of FSLoRA in more complex models. Specifically, we fine-tune the LLaMA-7B model on the Wizard, Dolly-15k, and Alpaca datasets and evaluate it on 1444 MMLU samples (available at: https://github.com/ATP-1010/FederatedLLM). For Wizard and Dolly-15k, we adopt the same heterogeneous data partitioning as (Wang et al., 2024). Since the Alpaca dataset lacks a clear task or domain structure, we apply a uniform random partitioning strategy to distribute the data across clients. We tune the q_proj and v_proj modules and set the local LoRA ranks $k_i = [64, 32, 16, 16, 8, 8, 4, 4, 4, 4]$ for 10 clients. The parameter settings are aligned with those in (Wang et al., 2024).

Table G.2: Performance comparison on LLaMA-7B model.

| Method | Wizard | Dolly-15k | Alpaca | Avg |
|---|---|---|---|---|
| HeteroLoRA | 27.15 | 26.70 | 28.74 | 27.53 |
| FlexLoRA | 28.25 | 35.60 | 30.40 | 31.42 |
| FLoRA | 27.91 | 28.50 | 29.54 | 28.65 |
| FSLoRA | 30.33 | 40.79 | 30.68 | 33.93 |

As shown in Table G.2, FSLoRA achieves the highest average performance across all three datasets compared to baselines. These results demonstrate FSLoRA's potential for effective fine-tuning with the large-scale LLaMA-7B model under heterogeneous client settings.

## H    FURTHER EXPERIMENTS

In this section, we provide additional results, including detailed per-task comparisons from the commonsense reasoning benchmark corresponding to Figures 3(a) and 3(b). In addition, we further investigate the impact of the number of local updates $H$ on the convergence, the robustness of FSLoRA under dynamic sketching ratio, and the integration of communication compression and sketching.

### H.1    FURTHER DETAILS ON THE ABLATION STUDY

**Impact of sketching:** In Figure 5, we compare the performance of FSLoRA with and without sketching on eight tasks from the commonsense reasoning benchmark using the LLaMA-3.2-3B model. Notably, FSLoRA without sketching is equivalent to the vanilla Federated LoRA. For FSLoRA with sketching, we apply a uniform sketching ratio of $k_i/r = 0.5$ across all distributed clients. The uploading budget for each client is set to 200 times the size of the full global LoRA modules at the corresponding rank. It is clear that FSLoRA with sketching consistently outperforms its non-sketched counterpart across these eight tasks, demonstrating the effectiveness of sketching in improving performance.

**Impact of the global rank:** In Figure 6, we present the impact of the rank of the global LoRA modules on FSLoRA's performance across eight tasks from the commonsense reasoning benchmark. We consider four configurations: 1) $r = 8$, $k_i/r = 1$, 2) $r = 16$, $k_i/r = 0.5$, 3) $r = 32$, $k_i/r = 0.25$, and 4) $r = 64$, $k_i/r = 0.125$. The rank of submatrices updated by the clients at each iteration remains consistent across all configurations (i.e., $k_i = 8$), ensuring that the communication and computational resources on the client side are kept fixed for all cases. In the ARC-Easy task, performance decreases as the rank increases to 64, potentially due to overfitting. In general, FSLoRA shows improved performance as the rank increases.

### H.2    IMPACT OF LOCAL UPDATES

Based on the commonsense reasoning benchmark and the LLaMA-3.2-3B model, we evaluated the convergence behavior of FSLoRA under varying numbers of local updates (i.e., $H$). The experimental results are presented in Figure 7. In the low-to-moderate regime of local updates (i.e., $H \in 10, 20, 100$), FSLoRA demonstrates a clear acceleration in convergence as $H$ increases. For example, moving from $H = 10$ to $H = 20$ substantially reduces the number of communication rounds required to reach the same level of testing accuracy, while further increasing $H$ to 100 yields even faster progress toward convergence. This observation indicates that a moderate increase in local updates allows clients to improve communication efficiency. However, when the number of local updates is pushed beyond this range (e.g., $H = 200$), no additional convergence gain is observed. These findings align well with our theoretical analysis in Section 4, which shows that FSLoRA can achieve a convergence speedup under certain conditions on $H$.

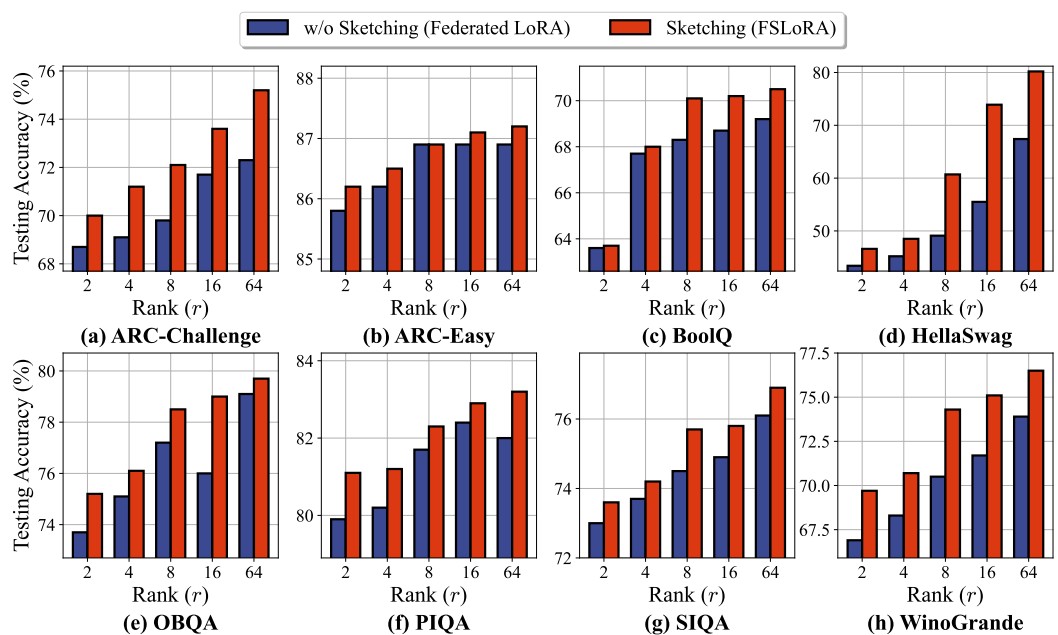

Figure 5: Comparison of FSLoRA with and without sketching, with an upload budget $200\times$ the global LoRA module size at each rank. This is based on the commonsense reasoning benchmark and the LLaMA-3.2-3B model. We observe that the sketching mechanism improves performance across all considered tasks. The average accuracy of the eight tasks is shown in Figure 3(a).

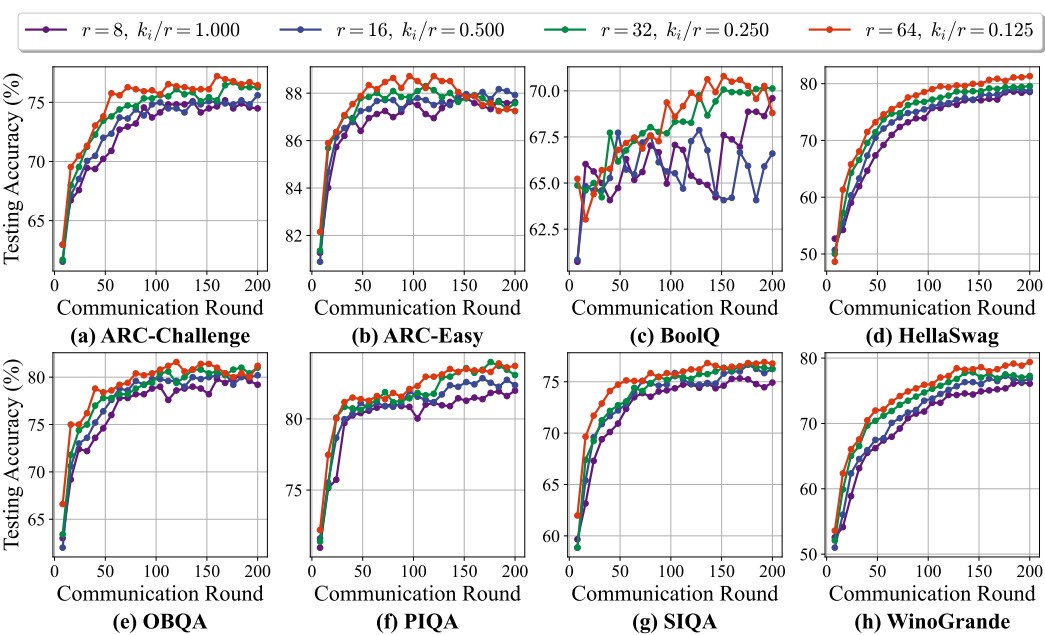

Figure 6: Impact of the rank of global LoRA modules on FSLoRA, given a fixed rank for the updated submatrices at the clients. This is based on the commonsense reasoning benchmark and the LLaMA-3.2-3B model. Overall, FSLoRA demonstrates improved performance as the global rank increases. The average accuracy of the eight tasks is shown in Figure 3(b).

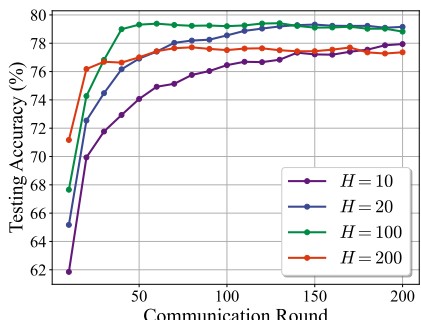

Figure 7: Impact of the number of local updates on FSLoRA's convergence. This is based on the commonsense reasoning benchmark and the LLaMA-3.2-3B model. In a certain range, i.e., from 10 to 100, FSLoRA achieves a fast convergence as $H$ increases.

### H.3 DYNAMIC SKETCHING RATIOS

While our primary focus is on developing a heterogeneous federated LoRA method under a standard static setup, following prior works (Wang et al., 2024; Cho et al., 2024; Bai et al., 2024), the proposed FSLoRA algorithm can be naturally extended to dynamic, time-varying resource environments. The modification is straightforward: we allow the sparsity levels, corresponding to the sketching ratios of FSLoRA, of the matrices in the sketching set $\mathcal{S}_i$ in Algorithm 1 to become time-varying, while keeping the remaining steps unchanged.

We empirically validate the effectiveness of FLoRA under this dynamic setting. In the simulation, we group clients into three capability levels *low*, *medium*, and *high*, assigned sketching ratio ranges $[0.125, 0.25]$, $[0.25, 0.5]$, and $[0.5, 1.0]$, respectively, to balance local training latencies across groups. Within each range, the sketching ratios are allowed to vary dynamically. The results, reported in Table H.1, show that FSLoRA maintains comparable performance when moving from the static to the dynamic case, demonstrating its robustness under time-varying sketching ratios.

Table H.1: The performance of FSLoRA under static and dynamic sketching ratios. This is based on the commonsense reasoning benchmark and the LLaMA-3.2-3B model. FSLoRA maintains comparable performance when moving from the static to the dynamic case.

| Ratios | ARC-c | ARC-e | BoolQ | HSwag | OBQA | PIQA | SIQA | Wino | Avg. |
|--------|-------|-------|-------|-------|------|------|------|------|------|
| Static | 76.1 | 87.1 | 70.0 | 81.7 | 81.4 | 82.6 | 76.4 | 78.9 | 79.3 |
| Dynamic | 75.5 | 87.7 | 69.2 | 81.3 | 81.2 | 82.2 | 76.0 | 78.8 | 79.0 |

### H.4 INTEGRATION OF SKETCHING AND TOP-K COMPRESSION

Building on the commonsense reasoning benchmark and the LLaMA-3.2-3B model, we further explore the integration of two orthogonal techniques, sketching and top-k compression, to further reduce the uplink communication overhead of clients in FSLoRA.

Specifically, with sketching, each client activates and updates submatrices of the global LoRA weights, $[\mathbf{b}_j]_{j \in \mathcal{I}_i}, [\mathbf{a}_j^\top]_{j \in \mathcal{I}_i}$, which are selected at the beginning of each round:

$$\mathbf{B}\mathbf{S}_i\mathbf{A} = \sum_{j \in \mathcal{I}_i} \frac{r}{k_i} \mathbf{b}_j \mathbf{a}_j^\top = \frac{r}{k_i} [\mathbf{b}_j]_{j \in \mathcal{I}_i} [\mathbf{a}_j^\top]_{j \in \mathcal{I}_i}, \tag{19}$$

where $\mathbf{b}_j$ and $\mathbf{a}_j^\top$ denote the $j$-th column of module $\mathbf{B}$ and the $j$-th row of module $\mathbf{A}$, respectively. By limiting updates to submatrices $[\mathbf{b}_j]_{j \in \mathcal{I}_i}$ and $[\mathbf{a}_j^\top]_{j \in \mathcal{I}_i}$, FSLoRA reduces communication and computation. To further reduce communication cost, we can apply Top-$k$ compression to the uploading stage. For instance, instead of sending the full $\Delta[\mathbf{b}_j]_{j \in \mathcal{I}_i}$, each client transmits the compressed update $\text{Top}_k(\Delta[\mathbf{b}_j]_{j \in \mathcal{I}_i})$. Sketching selects the update submatrix at the beginning of

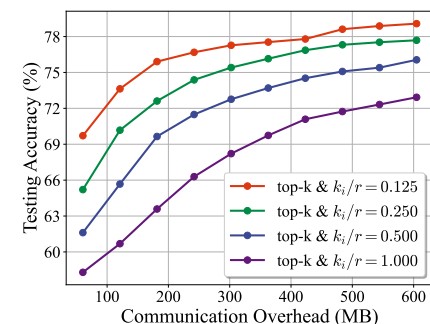

Figure 8: Testing accuracy versus communication overhead using float 32 precision. Lower sketching ratios achieve higher accuracy at the same communication cost, demonstrating that combining sketching with top-$k$ compression leads to more communication-efficient training.

each round, while compression further reduces its transmission cost at the uploading stage. These two techniques operate at different stages and can jointly improve communication efficiency.

In our setup, the compression ratio is fixed at $0.5$ for all methods, while the sketching ratio $k_i/r$ varies over $\{0.125, 0.25, 0.5, 1\}$. Notably, FSLoRA with sketching ratio $k_i/r = 1$ corresponds to the vanilla Federated LoRA (i.e., without sketching). Figure 8 plots testing accuracy versus communication overhead, where the x-axis represents the amount of data uploaded per client (in MB), assuming parameters are stored in float 32 precision. The results clearly show that integrating sketching with top-k compression further improves communication efficiency: methods with lower sketching ratios consistently achieve higher accuracy under the same communication budget, highlighting the potential of FSLoRA for scalable and communication-efficient collaborative LLM fine-tuning.

# I  IMPLEMENTATION DETAILS

## I.1  DETAILS ON HYPERPARAMETERS

Unless stated otherwise, the hyperparameters used in this work are as follows.

Table I.1: The hyperparameters for RoBERTa & GLUE and LLaMA-3.2-3B & commonsense reasoning benchmarks.

| Hyperparameter | RoBERTa & GLUE | LLaMA-3.2-3B & commonsense reasoning |
|---|---|---|
| Dirichlet parameter | 0.1 | 0.1 |
| Batch size | 16 | 16 |
| LoRA dropout rate | 0.1 | 0.1 |
| Learning rate, $\gamma$ | 5e-4 | 3e-4 |
| Communication round, $T$ | 200 | 200 |
| Local iteration number, $H$ | 50 | 20 |
| Target module | ["query", "value", "classification head"] | ["q_proj", "k_proj", "v_proj", "up_proj", "down_proj"] |

## I.2  DETAILS ON DATASETS

### I.2.1  GLUE BENCHMARK

GLUE is a widely recognized benchmark designed to assess the natural language understanding capabilities of language models (Wang, 2018).

- **CoLA** focuses on whether a given sentence is acceptable according to linguistic rules. It evaluates a model's ability to recognize well-formed sentences.
    - ▷ Input: A single sentence.
    - ☆ Output: A label indicating whether the sentence is acceptable or unacceptable.

- **SST-2** is designed for sentiment classification on movie reviews or short texts. It tests whether a model can correctly identify positive or negative sentiment in a given sentence.
    - ▷ Input: A single sentence.
    - ☆ Output: A label indicating positive or negative sentiment.
- **MRPC** checks if two sentences are paraphrases of each other, i.e., if they mean the same thing.
    - ▷ Input: Two sentences ('sentence1' and 'sentence2').
    - ☆ Output: A label indicating either equivalent or not equivalent.
- **QQP** tests a model's ability to determine if two questions ask the same thing.
    - ▷ Input: Two questions.
    - ☆ Output: A label indicating duplicate or not duplicate.
- **MNLI** tests whether a given hypothesis is entailed, contradicted, or neutral with respect to a premise.
    - ▷ Input: A premise (first sentence) and a hypothesis (second sentence).
    - ☆ Output: A label indicating entailment, contradiction, or neutral.
- **QNLI** aims to determine if a context sentence correctly answers a given question.
    - ▷ Input: A question and a sentence.
    - ☆ Output: A label indicating the sentence answers the question or it does not.
- **RTE** provides pairs of sentences to see if one implies the other.
    - ▷ Input: Two sentences ('sentence1' and 'sentence2')
    - ☆ Output: A label indicating whether the meaning of one sentence is entailed from the other one.

### I.2.2 COMMONSENSE REASONING BENCHMARK

The training set of the commonsense reasoning benchmark is a mixture of multiple datasets including about 170K training samples from ARC-c/e (Clark et al., 2018), BoolQ (Clark et al., 2019), HellaSwag (Zellers et al., 2019), OBQA (Mihaylov et al., 2018), PIQA (Bisk et al., 2020), SIQA (Sap et al., 2019), and WinoGrande (Sakaguchi et al., 2021) datasets.

- **ARC-c/e** contains the challenge and easy question set from the ARC dataset of genuine grade-school level, multiple-choice science questions.

- **BoolQ** is a question-answering dataset with yes/no questions derived from natural, real-world scenarios.

- **HellaSwag** includes questions for commonsense natural language inference, where a context and multiple endings are given, requiring the most coherent ending to be selected.

- **OBQA** involves multi-step problem-solving that combines commonsense knowledge, reasoning, and comprehension of accompanying textual information.

- **PIQA** focuses on questions requiring physical commonsense to solve. Each question offers two answer choices.

- **SIQA** targets reasoning about human actions and their social implication.

- **WinoGrande** is designed as a binary-choice fill-in-the-blank task, this dataset evaluates the ability to resolve ambiguous sentences through commonsense reasoning.

The input template, i.e., prompt format for these datasets is detailed in Table I.2.

Table I.2: The prompt template of the commonsense reasoning datasets (Hu et al., 2023).

| Dataset | Input Template |
|---|---|
| ARC-c/e | Please choose the correct answer to the question: [QUESTION]
Answer1: [ANSWER_1]
Answer2: [ANSWER_2]
Answer3: [ANSWER_3]
Answer4: [ANSWER_4]
Answer format: answer1/answer2/answer3/answer4
the correct answer is [ANSWER] |
| BoolQ | Please answer the following question with true or false, question: [QUESTION]
Answer format: true/false
the correct answer is [ANSWER] |
| HellaSwag | Please choose the correct ending to complete the given sentence: [ACTIVITY_LABEL]: [CONTEXT]
Ending1: [ENDING_1]
Ending2: [ENDING_2]
Ending3: [ENDING_3]
Ending4: [ENDING_4]
Answer format: ending1/ending2/ending3/ending4
the correct answer is [ANSWER] |
| OBQA | Please choose the correct answer to the question: [QUESTION]
Answer1: [ANSWER_1]
Answer2: [ANSWER_2]
Answer3: [ANSWER_3]
Answer4: [ANSWER_4]
Answer format: answer1/answer2/answer3/answer4
the correct answer is [ANSWER] |
| PIQA | Please choose the correct solution to the question: [QUESTION]
Solution1: [SOLUTION_1]
Solution2: [SOLUTION_2]
Answer format: solution1/solution2
the correct answer is [ANSWER] |
| SIQA | Please choose the correct answer to the question: [QUESTION]
Answer1: [ANSWER_1]
Answer2: [ANSWER_2]
Answer3: [ANSWER_3]
Answer format: answer1/answer2/answer3
the correct answer is [ANSWER] |
| WinoGrande | Please choose the correct answer to fill in the blank to complete the given sentence: [SENTENCE]
Option1: [OPTION_1]
Option2: [OPTION_2]
the correct answer is [ANSWER] |

# J PROOF OF THE THEORETICAL RESULTS

## J.1 PRELIMINARIES

Before presenting the proof of the main results, we first introduce some preliminary facts that will be used later. Throughout this work, $\| \cdot \|$ denotes the Frobenius norm when applied to a matrix and the $\ell_2$ norm when applied to a vector.

**Lemma J.1.** *Suppose a sequence of independent random matrices $\{\mathbf{P}_i\}_{i=1}^N$ satisfy $\mathbb{E}[\mathbf{P}_i] = \mathbf{0}, \forall i$. Then,*

$$\mathbb{E}\left\|\frac{1}{N}\sum_{i=1}^N \mathbf{P}_i\right\|^2 = \frac{1}{N^2}\sum_{i=1}^N \mathbb{E}\|\mathbf{P}_i\|^2. \tag{20}$$

**Lemma J.2.** *(Wang et al., 2021, Lemma 2) Suppose a sequence of random matrices $\{\mathbf{P}_i\}_{i=1}^N$ satisfy $\mathbb{E}\left[\mathbf{P}_i \mid \mathbf{P}_{i-1},\mathbf{P}_{i-2},\ldots,\mathbf{P}_1\right] = \mathbf{0}, \forall i$. Then,*

$$\mathbb{E}\left[\left\|\sum_{i=1}^N \mathbf{P}_i\right\|^2\right] = \sum_{i=1}^N \mathbb{E}\left[\|\mathbf{P}_i\|^2\right]. \tag{21}$$

**Lemma J.3.** *(Koloskova et al., 2020, Lemma 17) For any $a_0 \geq 0, b \geq 0, c \geq 0, d > 0$, there exist a constant $\eta \leq \frac{1}{d}$ such that*

$$\frac{a_0}{T\eta} + b\eta + c\eta^2 \leq 2\left(\frac{a_0 b}{T}\right)^{\frac{1}{2}} + 2c^{\frac{1}{3}}\left(\frac{a_0}{T}\right)^{\frac{2}{3}} + \frac{da_0}{T}. \tag{22}$$

**Lemma J.4** (Random sketching bounds)**.** *Let $\mathbf{S}$ be a random diagonal sketching matrix of the form*

$$\mathbf{S} = \frac{r}{k}\sum_{j\in\mathcal{I}} \mathbf{e}_j\,\mathbf{e}_j^\top, \tag{23}$$

*where $\mathbf{e}_1,\ldots,\mathbf{e}_r \in \mathbb{R}^r$ are standard unit basis vectors and $\mathcal{I} \subseteq \{1,\ldots,r\}$ is chosen uniformly at random with $|\mathcal{I}| = k$. Then any matrix $\mathbf{X}$ we have*

$$\|\mathbf{X}\,\mathbf{S}\|^2 \leq \frac{r^2}{k^2}\|\mathbf{X}\|^2, \tag{24}$$

*and in expectation we have*

$$\mathbb{E}_\mathbf{S}\left[\|\mathbf{X}\,\mathbf{S}\|^2\right] \leq \frac{r}{k}\|\mathbf{X}\|^2. \tag{25}$$

*Proof.* Since $\mathbf{S}$ is diagonal with exactly $k$ diagonal entries equal to $\frac{r}{k}$ and the rest zero, its largest eigenvalue is $\frac{r}{k}$. Squaring gives

$$\mathbf{S}\,\mathbf{S}^\top = \mathbf{S}^2 \preceq \frac{r^2}{k^2}\mathbf{I}, \tag{26}$$

Equivalently,

$$\mathbf{x}^\top\left(\mathbf{S}\,\mathbf{S}^\top\right)\mathbf{x} \leq \frac{r^2}{k^2}\|\mathbf{x}\|^2, \forall\mathbf{x}. \tag{27}$$

Setting $\mathbf{x} = \mathbf{x}_j$ to be the $j$-th column of $\mathbf{X}$ and summing over $j$ implies

$$\|\mathbf{X}\,\mathbf{S}\|^2 = \sum_j \|\mathbf{S}^\top\mathbf{x}_j\|^2 = \sum_j \mathbf{x}_j^\top\left(\mathbf{S}\,\mathbf{S}^\top\right)\mathbf{x}_j \leq \frac{r^2}{k^2}\sum_j \|\mathbf{x}_j\|^2 = \frac{r^2}{k^2}\|\mathbf{X}\|^2, \tag{28}$$

which proves (24).

For the expected bound (25), note that each diagonal index $j \in \{1,\ldots,r\}$ is included in $\mathcal{I}$ with probability $\frac{k}{r}$. Hence the expectation of $\mathbf{S}^2$ satisfies

$$\mathbb{E}_\mathbf{S}\left[\mathbf{S}^2\right] = \frac{r^2}{k^2}\mathbb{E}\left[\sum_{j\in\mathcal{I}}\mathbf{e}_j\,\mathbf{e}_j^\top\right] = \frac{r^2}{k^2}\frac{k}{r}\mathbf{I} = \frac{r}{k}\mathbf{I}. \tag{29}$$

Thus for any vector $\mathbf{x}$,

$$\mathbb{E}_{\mathbf{S}}\Big[\|\mathbf{S}^\top \mathbf{x}\|^2\Big] \;=\; \mathbb{E}_{\mathbf{S}}\Big[\mathbf{x}^\top \mathbf{S}\,\mathbf{S}^\top \mathbf{x}\Big] \;=\; \mathbf{x}^\top\Big(\mathbb{E}[\mathbf{S}^2]\Big)\mathbf{x} \;=\; \frac{r}{k}\,\|\mathbf{x}\|^2. \tag{30}$$

Summing over columns of $\mathbf{X}$ again establishes

$$\mathbb{E}_{\mathbf{S}}\big[\|\mathbf{X}\,\mathbf{S}\|^2\big] \;=\; \sum_j \mathbb{E}_{\mathbf{S}}\big[\|\mathbf{S}^\top \mathbf{x}_j\|^2\big] \;=\; \sum_j \mathbf{x}_j^\top\Big(\mathbb{E}[\mathbf{S}^2]\Big)\mathbf{x}_j \;=\; \frac{r}{k}\,\|\mathbf{X}\|^2. \tag{31}$$

This completes the proof of Lemma J.4. $\qquad\qquad\square$

### J.2 PROOF OF LEMMA 3.2

From the chain rule for matrix calculus, we know that:

$$\nabla_{\mathbf{Y}} g(\mathbf{XY}) = \mathbf{X}^\top \nabla g(\mathbf{XY}), \; \nabla_{\mathbf{X}} g(\mathbf{XY}) = \nabla g(\mathbf{XY})\mathbf{Y}^\top, \tag{32}$$

where $\nabla g(\mathbf{XY})$ denotes the gradient of $g$ to $\mathbf{XY}$. Applying this to $\ell(\mathbf{W}_0 + \mathbf{BSA}, \xi)$, we proceed as follows:
To compute the gradient with respect to $\mathbf{B}$, set $\mathbf{X} = \mathbf{B}$ and $\mathbf{Y} = \mathbf{SA}$:

$$\nabla_{\mathbf{B}}\ell(\mathbf{W}_0 + \mathbf{BSA}, \xi) = \nabla\ell(\mathbf{W}_0 + \mathbf{BSA}, \xi)(\mathbf{SA})^\top. \tag{33}$$

Similarly, to compute the gradient with respect to $\mathbf{A}$, set $\mathbf{X} = \mathbf{BS}$ and $\mathbf{Y} = \mathbf{A}$:

$$\nabla_{\mathbf{A}}\ell(\mathbf{W}_0 + \mathbf{BSA}, \xi) = \mathbf{S}^\top\mathbf{B}^\top\nabla\ell(\mathbf{W}_0 + \mathbf{BSA}, \xi). \tag{34}$$

### J.3 PROOF OF THEOREM 4.4

The proof of Theorem 4.4 relies on the following proposition.

**Proposition J.5.** *Under Assumption 4.1,* $\tilde{f}_i(\mathbf{X}; \mathbf{S}) \;=\; f_i(\mathbf{BS}, \mathbf{A})$, $\mathbf{S} \;\in\; \mathcal{S}_i$, $f_i^{\mathcal{S}}(\mathbf{X}) \;=\; \mathbb{E}_{\mathbf{S}\sim\mathcal{S}_i}[\tilde{f}_i(\mathbf{X}; \mathbf{S})]$, *and* $f^{\mathcal{S}}(\mathbf{X}) \;=\; \frac{1}{N}\sum_{i=1}^N f_i^{\mathcal{S}}(\mathbf{X})$ *are smooth with parameters* $L\frac{r^2}{k_i^2}$, $L\frac{r}{k_i}$, *and* $\left(\frac{1}{N}\sum_{i=1}^N \frac{r}{k_i}\right) L$, *respectively.*

The proof of Proposition J.5 is deferred to Appendix J.4. With this proposition, we are ready to prove Theorem 4.4.

In FSLoRA, the update direction in (5) corresponds to the negative stochastic gradient of $\ell(\mathbf{W}_0 + \mathbf{BSA}, \xi)$ with respect to $[\mathbf{B}; \mathbf{A}]$ for a given sketch $\mathbf{S}_i^t$. We have defined $\tilde{\ell}(\mathbf{X}, \xi; \mathbf{S}) = \ell(\mathbf{W}_0 + \mathbf{BSA}, \xi)$. The iterative equation for the proposed FSLoRA algorithm thus can be written as

$$\mathbf{X}^{t+1} = \mathbf{X}^t - \gamma\frac{1}{N}\sum_{i=1}^N\sum_{h=0}^{H-1}\nabla_{\mathbf{X}}\tilde{\ell}(\mathbf{X}_i^{t,h}, \xi_i^{t,h}; \mathbf{S}_i^t), \tag{35}$$

where $\mathbf{g}_i^{t,h}$ denotes the stochastic gradient $\nabla_{\mathbf{X}}\tilde{\ell}(\mathbf{X}_i^{t,h}, \xi_i^{t,h}; \mathbf{S}_i^t)$. Based on the smoothness of $f^{\mathcal{S}}(\mathbf{X})$, i.e., Proposition J.5, we have

$$\mathbb{E}[f^{\mathcal{S}}(\mathbf{X}^{t+1})] \le \mathbb{E}[f^{\mathcal{S}}(\mathbf{X}^t)] \underbrace{-\mathbb{E}\left\langle \nabla_{\mathbf{X}}f^{\mathcal{S}}(\mathbf{X}^t), \gamma\frac{1}{N}\sum_{i=1}^N\sum_{h=0}^{H-1}\mathbf{g}_i^{t,h}\right\rangle}_{T_1} + \underbrace{\frac{\gamma^2\bar{L}}{2}\mathbb{E}\left\|\frac{1}{N}\sum_{i=1}^N\sum_{h=0}^{H-1}\mathbf{g}_i^{t,h}\right\|^2}_{T_2},$$
$$\tag{36}$$

where $\bar{L} = \left(\frac{1}{N}\sum_{i=1}^N\frac{r}{k_i}\right) L$.

For $T_1$, we have

$$T_1 = - H\mathbb{E}\left\langle \nabla_{\mathbf{x}} f^{\mathcal{S}}(\mathbf{X}^t), \gamma \frac{1}{NH} \sum_{i=1}^{N} \sum_{h=0}^{H-1} \mathbf{g}_i^{t,h} \right\rangle$$

$$= - H\mathbb{E}\left\langle \nabla_{\mathbf{x}} f^{\mathcal{S}}(\mathbf{X}^t), \gamma \frac{1}{NH} \sum_{i=1}^{N} \sum_{h=0}^{H-1} \nabla_{\mathbf{x}} f_i^{\mathcal{S}}(\mathbf{X}_i^{t,h}) \right\rangle$$

$$= - \frac{\gamma H}{2}\mathbb{E}\left\| \nabla_{\mathbf{x}} f^{\mathcal{S}}(\mathbf{X}^t) \right\|^2 - \frac{\gamma H}{2}\mathbb{E}\left\| \frac{1}{NH} \sum_{i=1}^{N} \sum_{h=0}^{H-1} \nabla_{\mathbf{x}} f_i^{\mathcal{S}}(\mathbf{X}_i^{t,h}) \right\|^2$$

$$+ \frac{\gamma H}{2}\mathbb{E}\left\| \nabla_{\mathbf{x}} f^{\mathcal{S}}(\mathbf{X}^t) - \frac{1}{NH} \sum_{i=1}^{N} \sum_{h=0}^{H-1} \nabla_{\mathbf{x}} f_i^{\mathcal{S}}(\mathbf{X}_i^{t,h}) \right\|^2$$

$$\leq - \frac{\gamma H}{2}\mathbb{E}\left\| \nabla_{\mathbf{x}} f^{\mathcal{S}}(\mathbf{X}^t) \right\|^2 - \frac{\gamma H}{2}\mathbb{E}\left\| \frac{1}{N} \sum_{i=1}^{N} \sum_{h=0}^{H-1} \nabla_{\mathbf{x}} f_i^{\mathcal{S}}(\mathbf{X}_i^{t,h}) \right\|^2$$

$$+ \frac{\gamma}{2} \sum_{h=0}^{H-1} \mathbb{E}\left\| \frac{1}{N} \sum_{i=1}^{N} \nabla_{\mathbf{x}} f_i^{\mathcal{S}}(\mathbf{X}^t) - \frac{1}{N} \sum_{i=1}^{N} \nabla_{\mathbf{x}} f_i^{\mathcal{S}}(\mathbf{X}_i^{t,h}) \right\|^2$$

$$\leq - \frac{\gamma H}{2}\mathbb{E}\left\| \nabla_{\mathbf{x}} f^{\mathcal{S}}(\mathbf{X}^t) \right\|^2 - \frac{\gamma}{2H}\mathbb{E}\left\| \frac{1}{N} \sum_{i=1}^{N} \sum_{h=0}^{H-1} \nabla_{\mathbf{x}} f_i^{\mathcal{S}}(\mathbf{X}_i^{t,h}) \right\|^2$$

$$+ \frac{\gamma H L^2}{2} \frac{1}{NH} \sum_{i=1}^{N} \frac{r^2}{k_i^2} \sum_{h=0}^{H-1} \mathbb{E}\left\| \mathbf{X}_i^{t,h} - \mathbf{X}^t \right\|^2, \tag{37}$$

where the last inequalities follow Jensen's inequality and Proposition J.5.

For $T_2$, we have

$$T_2 = \mathbb{E}\left\| \frac{1}{N} \sum_{i=1}^{N} \sum_{h=0}^{H-1} (\mathbf{g}_i^{t,h} \mp \nabla_{\mathbf{x}} f_i^{\mathcal{S}}(\mathbf{X}_i^{t,h})) \right\|^2$$

$$\leq \frac{2}{N^2} \sum_{i=1}^{N} \mathbb{E}\left\| \sum_{h=0}^{H-1} (\mathbf{g}_i^{t,h} - \nabla_{\mathbf{x}} f_i^{\mathcal{S}}(\mathbf{X}_i^{t,h})) \right\|^2 + 2\mathbb{E}\left\| \frac{1}{N} \sum_{i=1}^{N} \sum_{h=0}^{H-1} \nabla_{\mathbf{x}} f_i^{\mathcal{S}}(\mathbf{X}_i^{t,h}) \right\|^2, \tag{38}$$

where the inequality follows the fact that $\mathbb{E}[\sum_{h=0}^{H-1} (\mathbf{g}_i^{t,h} - \nabla_{\mathbf{x}} f_i^{\mathcal{S}}(\mathbf{X}_i^{t,h}))] = 0$ and the independence between clients.

Furthermore, we bound the first term on the right-hand side of the above inequality as

$$\mathbb{E}\left\| \sum_{h=0}^{H-1} (\mathbf{g}_i^{t,h} - \nabla_{\mathbf{x}} f_i^{\mathcal{S}}(\mathbf{X}_i^{t,h})) \right\|^2 = \sum_{h=0}^{H-1} \mathbb{E}\left\| \mathbf{g}_i^{t,h} - \nabla_{\mathbf{x}} f_i^{\mathcal{S}}(\mathbf{X}_i^{t,h}) \right\|^2 \leq H\sigma^2 + \rho \sum_{h=0}^{H-1} \mathbb{E}\left\| \nabla_{\mathbf{x}} f_i^{\mathcal{S}}(\mathbf{X}_i^{t,h}) \right\|^2,$$

where the equality follows Lemma J.2 and the inequality follows Assumption 4.2. For $\left\| \nabla_{\mathbf{x}} f_i^{\mathcal{S}}(\mathbf{X}_i^{t,h}) \right\|^2$, utilizing Assumption 4.3 and Proposition J.5, we have

$$\left\| \nabla_{\mathbf{x}} f_i^{\mathcal{S}}(\mathbf{X}_i^{t,h}) \right\|^2 = \left\| \nabla_{\mathbf{x}} f_i^{\mathcal{S}}(\mathbf{X}_i^{t,h}) \mp \nabla_{\mathbf{x}} f_i^{\mathcal{S}}(\mathbf{X}^t) \mp \nabla_{\mathbf{x}} f^{\mathcal{S}}(\mathbf{X}^t) \right\|^2$$

$$\leq 3\left\| \nabla_{\mathbf{x}} f_i^{\mathcal{S}}(\mathbf{X}_i^{t,h}) - \nabla_{\mathbf{x}} f_i^{\mathcal{S}}(\mathbf{X}^t) \right\|^2 + 3\left\| \nabla_{\mathbf{x}} f_i^{\mathcal{S}}(\mathbf{X}^t) - \nabla_{\mathbf{x}} f^{\mathcal{S}}(\mathbf{X}^t) \right\|^2 + 3\left\| \nabla_{\mathbf{x}} f^{\mathcal{S}}(\mathbf{X}^t) \right\|^2$$

$$\leq 3\frac{r^2}{k_i^2} L^2 \left\| \mathbf{X}_i^{t,h} - \mathbf{X}^t \right\|^2 + 3(c_h + 1)\|\nabla_{\mathbf{x}} f^{\mathcal{S}}(\mathbf{X}^t)\|^2 + 3\rho\delta_h^2. \tag{39}$$

Combining the above three inequalities gives rise to

$$T_2 \leq \frac{2H}{N}(\sigma^2 + 3\rho\delta_h^2) + 2\mathbb{E}\left\|\frac{1}{N}\sum_{i=1}^{N}\sum_{h=0}^{H-1}\nabla_{\mathbf{x}}f_i^{\mathcal{S}}(\mathbf{X}_i^{t,h})\right\|^2 + \frac{6\rho(c_h+1)H}{N}\mathbb{E}\|\nabla_{\mathbf{x}}f^{\mathcal{S}}(\mathbf{X}^t)\|^2$$

$$+ \frac{6\rho H L^2}{N}T_3, \tag{40}$$

where $T_3 = \frac{1}{NH}\sum_{i=1}^{N}\frac{r^2}{k_i^2}\sum_{h=0}^{H-1}\mathbb{E}\left\|\mathbf{X}_i^{t,h} - \mathbf{X}^t\right\|^2$. Combining (36), (37), and (40) yields

$$\mathbb{E}[f^{\mathcal{S}}(\mathbf{X}^{t+1})] \leq \mathbb{E}[f^{\mathcal{S}}(\mathbf{X}^t)] - (\frac{\gamma H}{2} - 3\gamma^2\rho(c_h+1)\frac{H}{N}\bar{L})\mathbb{E}\left\|\nabla_{\mathbf{x}}f^{\mathcal{S}}(\mathbf{X}^t)\right\|^2 + \gamma^2\bar{L}\frac{H}{N}(\sigma^2 + 3\rho\sigma_h^2)$$

$$- (\frac{\gamma}{2H} - \gamma^2\bar{L})\mathbb{E}\left\|\frac{1}{N}\sum_{i=1}^{N}\sum_{h=0}^{H-1}\nabla_{\mathbf{x}}f_i^{\mathcal{S}}(\mathbf{X}_i^{t,h})\right\|^2 + (\frac{\gamma H L^2}{2} + 3\gamma^2\rho\bar{L}L^2\frac{H}{N})T_3, \tag{41}$$

where $\bar{L} = \left(\frac{1}{N}\sum_{i=1}^{N}\frac{r}{k_i}\right)L$. Let $\gamma \leq \min\{\frac{N}{24\rho(c_h+1)HL}, \frac{1}{2HL}, \frac{N}{6\rho L}\}$, we have

$$\mathbb{E}[f^{\mathcal{S}}(\mathbf{X}^{t+1})] \leq \mathbb{E}[f^{\mathcal{S}}(\mathbf{X}^t)] - \frac{3\gamma H}{8}\mathbb{E}\left\|\nabla_{\mathbf{x}}f^{\mathcal{S}}(\mathbf{X}^t)\right\|^2 + \gamma^2\bar{L}\frac{H}{N}(\sigma^2 + 3\rho\sigma_h^2) + \frac{5\gamma}{8}HL^2T_3. \tag{42}$$

For $T_3$, we have

$$T_3 = \frac{1}{NH}\sum_{i=1}^{N}\frac{r^2}{k_i^2}\sum_{h=0}^{H-1}\mathbb{E}\left\|\gamma\sum_{\tau=0}^{h-1}\mathbf{g}_i^{t,\tau}\right\|^2$$

$$= \gamma^2\frac{1}{NH}\sum_{i=1}^{N}\frac{r^2}{k_i^2}\sum_{h=0}^{H-1}\mathbb{E}\left\|\sum_{\tau=0}^{h-1}(\mathbf{g}_i^{t,\tau} \mp \nabla_{\mathbf{x}}f_i^{\mathcal{S}}(\mathbf{X}_i^{t,\tau}))\right\|^2$$

$$\leq 2\gamma^2\frac{1}{NH}\sum_{i=1}^{N}\frac{r^2}{k_i^2}\sum_{h=0}^{H-1}\sum_{\tau=0}^{h-1}\mathbb{E}\left\|\mathbf{g}_i^{t,\tau} - \nabla_{\mathbf{x}}f_i^{\mathcal{S}}(\mathbf{X}_i^{t,\tau})\right\|^2 + 2\gamma^2\frac{1}{NH}\sum_{i=1}^{N}\frac{r^2}{k_i^2}\sum_{h=0}^{H-1}h\sum_{\tau=0}^{h-1}\mathbb{E}\left\|\nabla_{\mathbf{x}}f_i^{\mathcal{S}}(\mathbf{X}_i^{t,\tau})\right\|^2$$

$$\leq 2\gamma^2H\sigma^2\left(\frac{1}{N}\sum_{i=1}^{N}\frac{r^2}{k_i^2}\right) + \frac{2\rho\gamma^2}{NH}\sum_{i=1}^{N}\frac{r^2}{k_i^2}\sum_{h=0}^{H-1}\sum_{\tau=0}^{h-1}\mathbb{E}\left\|\nabla_{\mathbf{x}}f_i^{\mathcal{S}}(\mathbf{X}_i^{t,\tau})\right\|^2$$

$$+ \frac{2\gamma^2}{NH}\sum_{i=1}^{N}\frac{r^2}{k_i^2}\sum_{h=0}^{H-1}h\sum_{\tau=0}^{h-1}\mathbb{E}\left\|\nabla_{\mathbf{x}}f_i^{\mathcal{S}}(\mathbf{X}_i^{t,\tau})\right\|^2$$

$$\leq 2\gamma^2H\sigma^2\left(\frac{1}{N}\sum_{i=1}^{N}\frac{r^2}{k_i^2}\right) + \frac{2(\rho+1)\gamma^2H}{N}\sum_{i=1}^{N}\frac{r^2}{k_i^2}\sum_{h=0}^{H-1}\mathbb{E}\left\|\nabla_{\mathbf{x}}f_i^{\mathcal{S}}(\mathbf{X}_i^{t,\tau})\right\|^2. \tag{43}$$

Plugging inequality (39) into inequality (43) yeilds

$$T_3 \leq 2\gamma^2H\left(\frac{1}{N}\sum_{i=1}^{N}\frac{r^2}{k_i^2}\right)\sigma^2 + 6(\rho+1)\gamma^2H^2\left(\frac{1}{N}\sum_{i=1}^{N}\frac{r^2}{k_i^2}\right)\sigma_h^2$$

$$+ 6(\rho+1)\gamma^2L^2H^2T_3 + 6(\rho+1)\left(\frac{1}{N}\sum_{i=1}^{N}\frac{r^2}{k_i^2}\right)(c_h+1)\gamma^2H^2\mathbb{E}\left\|\nabla_{\mathbf{x}}f^{\mathcal{S}}(\mathbf{X}^t)\right\|^2. \tag{44}$$

Denote $\kappa = \frac{1}{N}\sum_{i=1}^{N}\frac{r^2}{k_i^2}$, we simplify the above inequality as

$$(1 - 6(\rho+1)\gamma^2L^2H^2)T_3 \leq 2\kappa\gamma^2H^2(\sigma_g^2 + 3(\rho+1)\sigma_h^2) + 6\kappa(\rho+1)(c_h+1)\gamma^2H^2\mathbb{E}\left\|\nabla_{\mathbf{x}}f^{\mathcal{S}}(\mathbf{X}^t)\right\|^2.$$

Let $\gamma \leq \frac{1}{\sqrt{12(\rho+1)}HL}$, we get the bound for $T_3$

$$T_3 \leq 4\kappa\gamma^2H^2(\sigma^2 + 3(\rho+1)\sigma_h^2) + 12\kappa(\rho+1)(c_h+1)\gamma^2H^2\mathbb{E}\left\|\nabla_{\mathbf{x}}f^{\mathcal{S}}(\mathbf{X}^t)\right\|^2. \tag{45}$$

Plugging the bound for $T_3$ into inequality (42) gives rise to

$$\mathbb{E}[f^{\mathcal{S}}(\mathbf{X}^{t+1})] \leq \mathbb{E}[f^{\mathcal{S}}(\mathbf{X}^t)] - (\frac{3\gamma H}{8} - \frac{5\gamma H}{8} L^2 \left(12\kappa(\rho+1)(c_h+1)\gamma^2 H^2\right)) \mathbb{E}\left\|\nabla_{\mathbf{X}} f^{\mathcal{S}}(\mathbf{X}^t)\right\|^2$$
$$+ \gamma^2 \bar{L} \frac{H}{N}(\sigma^2 + 3\rho\sigma_h^2) + \frac{5\gamma}{8} HL^2 \cdot 4\kappa\gamma^2 H^2(\sigma^2 + 3(\rho+1)\sigma_h^2). \quad (46)$$

Let $\gamma \leq \frac{1}{8\sqrt{\kappa(\rho+1)(c_h+1)}HL}$, we obtain

$$\mathbb{E}[f^{\mathcal{S}}(\mathbf{X}^{t+1})] \leq \mathbb{E}[f^{\mathcal{S}}(\mathbf{X}^t)] - \frac{\gamma H}{4}\mathbb{E}\left\|\nabla_{\mathbf{X}} f^{\mathcal{S}}(\mathbf{X}^t)\right\|^2 + \gamma^2 \bar{L}\frac{H}{N}\sigma_\rho^2 + \frac{5}{2}\kappa\gamma^3 H^3 L^2 \sigma_\rho^2, \quad (47)$$

where $\sigma_\rho^2 = \sigma^2 + 3(\rho+1)\sigma_h^2$.

Telescoping the above inequality from $t = 0$ to $T - 1$, we have

$$\frac{1}{T}\sum_{t=0}^{T-1}\mathbb{E}\left\|\nabla_{\mathbf{X}} f^{\mathcal{S}}(\mathbf{X}^t)\right\|^2 \leq 4\frac{f^{\mathcal{S}}(\mathbf{X}^0) - f^*}{\gamma T H} + \gamma\frac{4\bar{L}}{N}\sigma_\rho^2 + 10\gamma^2 H^2 \widetilde{L}L\sigma_\rho^2, \quad (48)$$

where $f^*$ denotes the lower bound of $f^{\mathcal{S}}(\mathbf{X})$ and $\widetilde{L} = \kappa L$.

Applying Lemma J.3 to the above inequality and letting $d = H$, it follows that there exists a learning rate $\gamma \leq \min\{\frac{N}{24\rho(c_h+1)H\bar{L}}, \frac{1}{8\sqrt{\widetilde{L}L(\rho+1)(c_h+1)}H}, \frac{1}{H}\}$ such that

$$\frac{1}{T}\sum_{t=0}^{T-1}\mathbb{E}\left\|\nabla_{\mathbf{X}} f^{\mathcal{S}}(\mathbf{X}^t)\right\|^2 \leq 8\frac{\sqrt{\bar{L}\mathcal{F}_0\sigma_\rho^2}}{\sqrt{NTH}} + 10(\widetilde{L}L)^{\frac{1}{3}}\left(\frac{\mathcal{F}_0\sigma_\rho}{T}\right)^{\frac{2}{3}} + \frac{4\mathcal{F}_0}{T}. \quad (49)$$

This completes the proof of Theorem 4.4.

## J.4 PROOF OF PROPOSITION J.5

i) For illustration, we need to recover $\mathbf{X}$ to $[\mathbf{B}; \mathbf{A}]$ in this proof. According to the definition of $\tilde{f}_i(\mathbf{X}; \mathbf{S})$ and $f_i(\mathbf{B}, \mathbf{A})$, we have

$$\tilde{f}_i(\mathbf{X}; \mathbf{S}) = \tilde{f}_i(\mathbf{B}, \mathbf{A}; \mathbf{S}) \quad (50)$$
$$= \mathbb{E}_{\xi\sim\mathcal{D}_i}[\ell(\mathbf{W}_0 + \mathbf{BSA}, \xi)]$$
$$= f_i(\mathbf{BS}, \mathbf{A}). \quad (51)$$

As $f_i(\mathbf{B}, \mathbf{A})$ is $L$-smooth, we have

$$f_i(\mathbf{BS} + \Delta\mathbf{BS}, \mathbf{A} + \Delta\mathbf{A}) \leq f_i(\mathbf{BS}, \mathbf{A}) + \left\langle \begin{bmatrix} \nabla_{\mathbf{BS}} f_i(\mathbf{BS}, \mathbf{A}) \\ \nabla_{\mathbf{A}} f_i(\mathbf{BS}, \mathbf{A}) \end{bmatrix}, \begin{bmatrix} \Delta\mathbf{BS} \\ \Delta\mathbf{A} \end{bmatrix} \right\rangle + \frac{L}{2}\left\|\begin{bmatrix} \Delta\mathbf{BS} \\ \Delta\mathbf{A} \end{bmatrix}\right\|^2. \quad (52)$$

According to (50) and (51), we have $\tilde{f}_i(\mathbf{B} + \Delta\mathbf{B}, \mathbf{A} + \Delta\mathbf{A}; \mathbf{S}) = f_i(\mathbf{BS} + \Delta\mathbf{BS}, \mathbf{A} + \Delta\mathbf{A})$ and $\tilde{f}_i(\mathbf{B}, \mathbf{A}; \mathbf{S}) = f_i(\mathbf{BS}, \mathbf{A})$. Combining these with (52) gives rise to

$$\tilde{f}_i(\mathbf{B} + \Delta\mathbf{B}, \mathbf{A} + \Delta\mathbf{A}; \mathbf{S}) \leq \tilde{f}_i(\mathbf{B}, \mathbf{A}; \mathbf{S}) + \left\langle \begin{bmatrix} \nabla_{\mathbf{BS}} f_i(\mathbf{BS}, \mathbf{A}) \\ \nabla_{\mathbf{A}} f_i(\mathbf{BS}, \mathbf{A}) \end{bmatrix}, \begin{bmatrix} \Delta\mathbf{BS} \\ \Delta\mathbf{A} \end{bmatrix} \right\rangle + \frac{L}{2}\left\|\begin{bmatrix} \Delta\mathbf{BS} \\ \Delta\mathbf{A} \end{bmatrix}\right\|^2. \quad (53)$$

We denote

$$L(\mathbf{W}_0 + \mathbf{BSA}) = \tilde{f}_i(\mathbf{B}, \mathbf{A}; \mathbf{S}) = \mathbb{E}_{\xi\sim\mathcal{D}_i}[\ell(\mathbf{W}_0 + \mathbf{BSA}, \xi)]. \quad (54)$$

Note that $\nabla_{\mathbf{BS}} f_i(\mathbf{BS}, \mathbf{A}) = \nabla L(\mathbf{W}_0 + \mathbf{BSA})\mathbf{A}^\top$ and $\nabla_{\mathbf{A}} f_i(\mathbf{BS}, \mathbf{A}) = \mathbf{S}^\top \mathbf{B}^\top \nabla L(\mathbf{W}_0 + \mathbf{BSA})$. We thus have

$$\left\langle \begin{bmatrix} \nabla_{\mathbf{BS}} f_i(\mathbf{BS}; \mathbf{A}) \\ \nabla_{\mathbf{A}} f_i(\mathbf{BS}; \mathbf{A}) \end{bmatrix}, \begin{bmatrix} \Delta\mathbf{BS} \\ \Delta\mathbf{A} \end{bmatrix} \right\rangle = \left\langle \begin{bmatrix} \nabla L(\mathbf{W}_0 + \mathbf{BSA})\mathbf{A}^\top \\ \mathbf{S}^\top\mathbf{B}^\top\nabla L(\mathbf{W_0} + \mathbf{BSA}) \end{bmatrix}, \begin{bmatrix} \Delta\mathbf{BS} \\ \Delta\mathbf{A} \end{bmatrix} \right\rangle$$
$$= \left\langle \begin{bmatrix} \nabla L(\mathbf{W}_0 + \mathbf{BSA})\mathbf{A}^\top\mathbf{S}^\top \\ \mathbf{S}^\top\mathbf{B}^\top\nabla L(\mathbf{W_0} + \mathbf{BSA}) \end{bmatrix}, \begin{bmatrix} \Delta\mathbf{B} \\ \Delta\mathbf{A} \end{bmatrix} \right\rangle$$
$$= \left\langle \begin{bmatrix} \nabla_{\mathbf{B}}\tilde{f}_i(\mathbf{B}, \mathbf{A}; \mathbf{S}) \\ \nabla_{\mathbf{A}}\tilde{f}_i(\mathbf{B}, \mathbf{A}; \mathbf{S}) \end{bmatrix}, \begin{bmatrix} \Delta\mathbf{B} \\ \Delta\mathbf{A} \end{bmatrix} \right\rangle, \quad (55)$$

where the last equality follows the fact that $\tilde{f}_i(\mathbf{B}, \mathbf{A}; \mathbf{S}) = L(\mathbf{W}_0 + \mathbf{BSA})$ defined in (54) and

$$
\begin{bmatrix} \nabla_{\mathbf{B}} \tilde{f}_i(\mathbf{B}, \mathbf{A}; \mathbf{S}) \\ \nabla_{\mathbf{A}} \tilde{f}_i(\mathbf{B}, \mathbf{A}; \mathbf{S}) \end{bmatrix} = \begin{bmatrix} \nabla L(\mathbf{W}_0 + \mathbf{BSA}) \mathbf{A}^\top \mathbf{S}^\top \\ \mathbf{S}^\top \mathbf{B}^\top \nabla L(\mathbf{W}_0 + \mathbf{BSA}) \end{bmatrix}. \tag{56}
$$

Plugging (55) into (53) gives rise to

$$
\tilde{f}_i(\mathbf{B} + \Delta\mathbf{B}, \mathbf{A} + \Delta\mathbf{A}; \mathbf{S}) \leq \tilde{f}_i(\mathbf{B}, \mathbf{A}; \mathbf{S}) + \left\langle \begin{bmatrix} \nabla_{\mathbf{B}} \tilde{f}_i(\mathbf{B}, \mathbf{A}; \mathbf{S}) \\ \nabla_{\mathbf{A}} \tilde{f}_i(\mathbf{B}, \mathbf{A}; \mathbf{S}) \end{bmatrix}, \begin{bmatrix} \Delta\mathbf{B} \\ \Delta\mathbf{A} \end{bmatrix} \right\rangle + \frac{L}{2} \left\| \begin{bmatrix} \Delta\mathbf{BS} \\ \Delta\mathbf{A} \end{bmatrix} \right\|^2. \tag{57}
$$

In particular, $\left\| \begin{bmatrix} \Delta\mathbf{BS} \\ \Delta\mathbf{A} \end{bmatrix} \right\|^2 = \|\Delta\mathbf{BS}\|^2 + \|\Delta\mathbf{A}\|^2$. From (24), we know $\|\Delta\mathbf{BS}\|^2 \leq \frac{r^2}{k_i^2}\|\Delta\mathbf{B}\|^2$. Therefore, we have $\left\| \begin{bmatrix} \Delta\mathbf{BS} \\ \Delta\mathbf{A} \end{bmatrix} \right\|^2 = \frac{r^2}{k_i^2} \left\| \begin{bmatrix} \Delta\mathbf{B} \\ \Delta\mathbf{A} \end{bmatrix} \right\|^2$. As a result, $\tilde{f}_i(\mathbf{B}, \mathbf{A}; \mathbf{S})$ (i.e., $\tilde{f}_i(\mathbf{X}, \mathbf{S})$) is $L\frac{r^2}{k_i^2}$-smooth.

ii) Note that $f_i^{\mathcal{S}}(\mathbf{X}) = \mathbb{E}_{\mathbf{S} \sim \mathcal{S}_i}[\tilde{f}_i(\mathbf{X}, \mathbf{S})]$. Therefore, we further take expectation for (57) over $\mathbf{S} \sim \mathcal{S}_i$, leading to

$$
f_i^{\mathcal{S}}(\mathbf{B} + \Delta\mathbf{B}, \mathbf{A} + \Delta\mathbf{A}) \leq f_i^{\mathcal{S}}(\mathbf{B}, \mathbf{A}) + \left\langle \begin{bmatrix} \nabla_{\mathbf{B}} f_i^{\mathcal{S}}(\mathbf{B}, \mathbf{A}) \\ \nabla_{\mathbf{A}} f_i^{\mathcal{S}}(\mathbf{B}, \mathbf{A}) \end{bmatrix}, \begin{bmatrix} \Delta\mathbf{B} \\ \Delta\mathbf{A} \end{bmatrix} \right\rangle + \frac{L}{2} \mathbb{E}_{\mathbf{S} \sim \mathcal{S}_i} \left\| \begin{bmatrix} \Delta\mathbf{BS} \\ \Delta\mathbf{A} \end{bmatrix} \right\|^2. \tag{58}
$$

In particular, $\mathbb{E}_{\mathbf{S} \sim \mathcal{S}_i} \left\| \begin{bmatrix} \Delta\mathbf{BS} \\ \Delta\mathbf{A} \end{bmatrix} \right\|^2 = \mathbb{E}_{\mathbf{S} \sim \mathcal{S}_i}\|\Delta\mathbf{BS}\|^2 + \|\Delta\mathbf{A}\|^2$. From (25), we know $\mathbb{E}_{\mathbf{S} \sim \mathcal{S}_i}\|\Delta\mathbf{BS}\|^2 \leq \frac{r}{k_i}\|\Delta\mathbf{B}\|^2$. In other words, $\mathbb{E}_{\mathbf{S} \sim \mathcal{S}_i} \left\| \begin{bmatrix} \Delta\mathbf{BS} \\ \Delta\mathbf{A} \end{bmatrix} \right\|^2 \leq \frac{r}{k_i} \left\| \begin{bmatrix} \Delta\mathbf{B} \\ \Delta\mathbf{A} \end{bmatrix} \right\|^2$. We thus claim that $f_i^{\mathcal{S}}(\mathbf{B}, \mathbf{A})$ (i.e., $f_i^{\mathcal{S}}(\mathbf{X})$) is $L\frac{r}{k_i}$-smooth.

iii) Finally, for $f^{\mathcal{S}}(\mathbf{X}) = \frac{1}{N}\sum_{i=1}^{N} f_i^{\mathcal{S}}(\mathbf{X})$, we have

$$
\nabla f^{\mathcal{S}}(\mathbf{X}) = \frac{1}{N}\sum_{i=1}^{N} \nabla f_i^{\mathcal{S}}(\mathbf{X}). \tag{59}
$$

Since $f_i^{\mathcal{S}}(\mathbf{X})$ is $L\frac{r}{k_i}$-smooth, we thus have

$$
\|\nabla f_i^{\mathcal{S}}(\mathbf{X}) - \nabla f_i^{\mathcal{S}}(\mathbf{Y})\| \leq L\frac{r}{k_i}\|\mathbf{X} - \mathbf{Y}\|, \quad \forall \mathbf{X}, \mathbf{Y}. \tag{60}
$$

To find the Lipschitz constant of $f^{\mathcal{S}}(\mathbf{X})$, we analyze the difference between the gradients at two points $\mathbf{X}$ and $\mathbf{Y}$:

$$
\begin{aligned}
\|\nabla f^{\mathcal{S}}(\mathbf{X}) - \nabla f^{\mathcal{S}}(\mathbf{Y})\| &= \left\| \frac{1}{N}\sum_{i=1}^{N} \left( \nabla f_i^{\mathcal{S}}(\mathbf{X}) - \nabla f_i^{\mathcal{S}}(\mathbf{Y}) \right) \right\| \\
&\leq \frac{1}{N}\sum_{i=1}^{N} \left\| \nabla f_i^{\mathcal{S}}(\mathbf{X}) - \nabla f_i^{\mathcal{S}}(\mathbf{Y}) \right\| \\
&\leq \left( \frac{1}{N}\sum_{i=1}^{N} \frac{r}{k_i}L \right) \|\mathbf{X} - \mathbf{Y}\|.
\end{aligned} \tag{61}
$$

Therefore, $f^{\mathcal{S}}(\mathbf{X})$ is $\left( \frac{1}{N}\sum_{i=1}^{N} \frac{r}{k_i}L \right)$-smooth.

