# OpenReview forum: "Federated Sketching LoRA: A Flexible Framework for Heterogeneous Collaborative Fine-Tuning of LLMs"
_ICLR.cc/2026/Conference — Submitted to ICLR 2026_

### Official Review · Reviewer_E2kV · 2025-10-30

**Soundness:** 3
**Presentation:** 4
**Contribution:** 2
**Rating:** 4
**Confidence:** 4

**Summary:**

The paper proposes Federated Sketching LoRA (FSLoRA), a heterogeneous federated fine-tuning framework for LLMs that keeps a large global LoRA rank on the server but lets each client update only a submatrix of the LoRA adapters via a random-k diagonal sketching matrix. By choosing client-specific sketching ratios ki​/r, FSLoRA reduces per-client compute/communication while retaining the expressivity of a higher global rank. The authors give a convergence analysis that quantifies how sketching ratios rescale smoothness constants and thus the convergence rate, and they report consistent accuracy gains vs. HeteroLoRA, FlexLoRA, and FLoRA and commonsense reasoning (LLaMA-3.2-3B), with comparable or lower GPU hours.

**Strengths:**

1. Random-k sketching gives unbiased submatrix updates with a clean convergence analysis that recovers FedAvg as ki→r.
2. Reduces both computation and communication without SVD or adapter merging.

**Weaknesses:**

1. FSLoRA adds a server-side duty each round to sample sketches, reconstruct sparse updates, and aggregate them.
2. Random-k is unbiased and makes the theory go through, but may discard consistently useful columns/rows. The authors test simple importance metrics and find them worse, partly because they break unbiasedness

**Questions:**

1. Your theory assumes random-k diagonal sketching with unbiasedness and yields scaled smoothness in the bound. In practice, random-k may repeatedly miss high-utility columns/rows. Can you extend the analysis to a data-dependent, non-uniform sketch policy？
2. You aggregate [B;A] and claim compatibility with secure aggregation, but there are no end-to-end systems measurements. How does this choice affect latency under many layers or clients？

---

> ### Author Response · Authors · 2025-11-18
> **Responses (1/2)**
>
> We thank the reviewer for the time, effort, and valuable feedback.
> ### **W1. Server-side duty**
> We want to clarify that among heterogeneous LoRA baselines, **FSLoRA guarantees convergence with almost minimal server-side overhead** (comparable to HeteroLoRA). The two extra server steps, **sampling sketches** and **reconstructing sparse updates**, are computationally light. Sampling requires generating a length-$r$ vector per client (much smaller than full LoRA modules), and reconstruction is essentially **simple padding**. Similar padding appears in HeteroLoRA, while FlexLoRA needs to compute LoRA products and perform SVD. FLoRA concatenates all LoRA modules and rebroadcasts them, incurring communication load that scales roughly linearly with the number of clients.
>
> For clarity, the table below summarizes server-side computation and communication across methods, where $N$ is the number of clients, $q$ is the downlink load of the global LoRA, and $k_i, r, m, n$ denote the local/global LoRA ranks and LoRA dimensions. A more detailed comparison can be found in Appendix A of the original manuscript.
>
> | Method | Server Computation | Server Communication (Downlink)  |
> | - | - | - |
> | HeteroLoRA | $\mathcal{O}(N r(m+n))$ | $q$ |
> | FlexLoRA | $\mathcal{O}((\sum_{i=1}^N k_i)mn))+\mathcal{O}(Nmn)+\mathcal{O}(\min\\{m, n\\}mn)$ | $q$ |
> | FLoRA  | $\mathcal{O}((\sum_{i=1}^N k_i)(m+n))$ | $\sum_{i=1}^{N} \frac{k_i}{r}q $ |
> | FSLoRA | $\mathcal{O}(N r(m+n))$ | $q(1+\frac{Nr}{32mn})$ |
>
> >In general, $q(1+\frac{Nr}{32mn}) \approx q$. The overal server-side complexity of FSLoRA is comparable to HeteroLoRA and more efficient than FlexLoRA and FLoRA.  In addition, we incorporated a numerical latency comparison in response to **Question 2**, which further demonstrates the efficiency of FSLoRA.
>
> ### **W2. Discard useful columns/rows**
> ***Random-k sketching guaranetees approximately uniform columns/rows exposure.*** LoRA parameterizes the update matrix as  $\mathbf{W} = \mathbf{BA} = b_1 a_1^\top + b_2 a_2^\top + \cdots + b_r a_r^\top$. In standard LoRA initialization, $\mathbf{B}$ is sampled with i.i.d. Gaussian entries and $\mathbf{A}$ is initialized to zero. This distribution is **invariant under permutations** of the $r$ columns of $\mathbf{B}$ and the corresponding rows of $\mathbf{A}$, so the rank-1 components $\\{b\_i a\_i^{\\top}\\}\_{i=1}^r$ are **statistically indistinguishable and have the same expected contribution** to $\mathbf{W}$.  The utility of individual columns or rows is generally difficult to define. Under random-k sketching, each rank-1 component in $\\{b\_i a\_i^{\\top}\\}\_{i=1}^r$ is sampled with equal probability per round. Over training, this ensures approximately **uniform exposure** to all components, **mirroring standard LoRA** where each column of $\mathbf{B}$ and row of $\mathbf{A}$ are updated at the same frequency. **As iterations increase, all components will eventually be fully trained.**
>
> ***Exploring importance based sketching.*** It remains an open question which metric is most appropriate for quantifying the importance of each LoRA column and row (even in the centralized LoRA training). Nevertheless, we investigated the spectral norm of each rank-1 component, $\\|b\_i a\_i^\top\\| = \\|b_i\\|\\|a_i\\|$, as a theoretically motivated proxy inspired by matrix analysis. We also evaluated a simpler alternative metric based on the magnitude sum, $\\|b_i\\| + \\|a_i\\|$. However, despite their intuitive appeal, both schemes consistently underperformed compared to the random-k selection, which yielded the strongest empirical results.
>
> To further verify the benefit of random- k selection, we also examine an alternative scheme that ensures unbiasedness by adjusting the diagonal elements of the sketching matrix. We set $s_i = \frac{1}{\pi_i}$ or 0 (not selected) and $\pi_i = \frac{\\|b_i\\|\\|a_i\\|}{\sum_{j=1}^r \\|b_i\\|\\|a_i\\|},$ where $\pi_i$ is the marginal probability that index $i$ is selected. With this construction, $\mathbb{E}[BSA] = B \mathbb{E}[S]A = BA,$ so the sketching remains **unbiased**. The table below reports the performance of FSLoRA under this unbiased importance-aware sketching scheme.
>
> |Method|ARC-c|ARC-e|BoolQ|HellaSwag|OBQA|PIQA|SIQA|WinoGrande| Avg.|
> | - | - | - | - | - | - | - | - | - |-|
> | Spectral nom | 71.9  | 86.5  | 55.2  | 75.4 | 73.4 | 81.1 | 72.5 | 69.7 | 73.2 |
> | Spectral nom (Unbiased ) | 73.4  | 86.2  | 60.5 | 77.9  | 75.8 | 81.6 | 74.6 | 73.1 | 75.4 |
> | Random-$k$ | 75.8  | 86.7  | 69.7  | 81.4 | 80.4 | 83.9 | 76.2 | 78.8 | 79.1 |
>
> As shown in the table, restoring unbiasedness slightly improves the performance, but it remains inferior to random-k sketching.
>
> > Our work demonstrates that random-k sketching achieves promising performance and comes with clean theoretical guarantees. We also explored several potential data-aware sketching, but none outperformed random-k in our evaluations. If the reviewer has further recommendations, we are happy to explore them.

---

> ### Author Response · Authors · 2025-11-18
> **Responses (2/2)**
>
> ### **Q1. Analysis extension**
> ***Data-dependent Analysis.*** To analyze convergence, the distributional properties of the sketching matrix is generally known in advance. This is consistent with existing works on sketching-based optimization: for example, [R1] establishes convergence under Gaussian sketching; [R2] uses sub-Gaussian sketches; and [R3] studies the randomized Kaczmarz sketch. In contrast, under a data-dependent sketching policy, the matrices $\mathbf{B}$ and $\mathbf{A}$ evolve dynamically during training, and their distributions are unknown. As a result, the distribution of the induced sketching matrix is not predetermined, making the characterization of its convergence properties an open question.
>
> ***Extension to non-uniform sketch.*** Assume each rank-1 component in $\\{b_1 a_1^\top, b_2 a_2^\top, \ldots, b_r a_r^\top\\}$ is associated with a static importance weight $\{w_1, w_2, \ldots, w_r\}$. We sample $k$ indices from $\{1,\ldots,r\}$ without replacement according to these weights, and denote the selected index set as $\mathcal{I}$. The diagonal entries of the sketching matrix $S$ are then defined as $s_i =
> \begin{cases}
> \dfrac{1}{\pi_i}, & i \in \mathcal{I}, \\\\
> 0, & i \notin \mathcal{I},
> \end{cases}
> $ where $\pi_i = \frac{k w_i}{\sum_{j=1}^r w_j}$ denotes the marginal probability that index $i$ is selected. With this construction,$\mathbb{E}[\mathbf{BSA}] = \mathbf{B}\mathbb{E}[S]\mathbf{A} = \mathbf{BA},$ so the sketching remains **unbiased**. Consequently, the convergence derivation in the original manuscript can be carried through with modified constants. In particular, the coefficients $\frac{r}{k_i}$ and $\frac{r^2}{k_i^2}$ in Lemma J.4 will be replaced by $\frac{\sum_{j=1}^r w_j}{k_i w_i}$ and $\left(\frac{\sum_{j=1}^r w_j}{k_i w_i}\right)^2$, respectively. Using unbiasedness  and modified Lemma J.4, the argument in Proposition J.5 remains valid after updating these constants. Consequently, we arrive at a bound analogous to Theorem 4.4, with the coefficients $\frac{r}{k_i}$ and $\frac{r^2}{k_i^2}$ replaced by $\frac{\sum_{j=1}^r w_j}{k_i w_i}$ and $\left(\frac{\sum_{j=1}^r w_j}{k_i w_i}\right)^2$. This completes the extension to the non-uniform sketch.
>
> [R1] "Iterative Double Sketching for Faster Least-Squares Optimization." *ICML 2022*.
>
> [R2] "Sketching for Distributed Deep Learning: A Sharper Analysis." *NeurIPS 2024*.
>
> [R3] "A randomized Kaczmarz algorithm with exponential convergence." *J. Fourier Anal. Appl.* 2009.
>
> > Although fully data-dependent sketching and its analysis remains to be an open problem, we would like to note that our work is the first to incorporate sketching into federated LoRA with a theoretical analysis, and it can be naturally extended to static non-uniform sketching.
>
> ### **Q2. Aggregation latency**
>
> Most of LoRA-based FL methods, including ours and existing baselines, have aggregation latency that grows linearly with the number of LoRA **layers**, since each layer is aggregated independently. The difference lies in how the methods scale with the number of clients: FlexLoRA requires the SVD of the average $\\frac{1}{N} \\sum\_{i=1}^N\\mathbf{B}\_i\\mathbf{A}\_i$, and FLoRA incurs downlink cost that grows linearly in the number of clients due to broadcasting $[\mathbf{B}\_1,\mathbf{B}\_2,\ldots,\mathbf{B}\_N]$ and $[\mathbf{A}\_1,\mathbf{A}\_2,\ldots,\mathbf{A}\_N]$. In contrast, FSLoRA (ours) reconstruct the local updates in the global size and aggregates $\\frac{1}{N} \\sum\_{i=1}^N \\mathbf{B}\_i$ and $\\frac{1}{N} \\sum\_{i=1}^N \\mathbf{A}\_i$, matching the complexity of HeteroLoRA. Therefore, our aggregation choice is efficient comparaed with baseline even if the number of clients increases. The detailed complexity comparison can be found in Appendix A of the original manuscript.
>
> **To address the concern about end-to-end latency**,  we compared the **per-round latency** of different aggregation strategies under different number of clients, accounting for local LoRA upload, server-side computation, and global LoRA dissemination. Server-side computation is measured under an A100 GPU, and uplink/downlink bandwidths are set to 5 MB/s and 10 MB/s. Latency (s) is reported for LLaMA-3.2-3B with local LoRA rank 16 (63 MB) and global rank 64 (252 MB).
>
> | Num clients | HeteroLoRA | FlexLoRA | FLoRA  | FSLoRA (Ours) |
> | ----------- | ---------- | -------- | ------ | ------------- |
> | 10          | 37.9       | 117.4    | 74.5   | 38.0          |
> | 50          | 38.2       | 119.5    | 322.8  | 38.5          |
> | 100         | 38.6       | 120.4    | 633.2  | 38.9          |
> | 200         | 39.4       | 124.6    | 1253.8 | 40.2          |
>
> > Both the complexity analysis (Appendix A) and the latency results here confirm the efficiency of FSLoRA’s aggregation.
>
> Again, we appreciate the reviewer’s time and thoughtful comments.

---

> ### Author Response · Authors · 2025-11-27
>
> Dear Reviewer E2kV,
>
> We hope this message finds you well.
>
> We want ensure we have addressed your concerns satisfactorily. If there are any additional points you'd like us to consider, please let us know. Your feedback is important to us, and we're eager to address any remaining concerns that you may have.
>
> Thank you for your time and effort in reviewing our work.
>
> Sincerely,
>
> The authors of Submission 12821

---

### Official Review · Reviewer_QhCG · 2025-10-31

**Soundness:** 3
**Presentation:** 3
**Contribution:** 3
**Rating:** 6
**Confidence:** 3

**Summary:**

This paper addresses the challenge of fine-tuning large language models on resource-limited clients, where data scarcity and heterogeneous computational capabilities pose significant obstacles. It proposed Federated Sketching LoRA (FSLoRA) to allow clients to update selected submatrices of global LoRA modules according to their resources, with adjustable sketching ratios to balance efficiency and performance. A rigorous convergence analysis is provided, and extensive experiments demonstrate consistent improvements over strong baselines across multiple datasets and LLMs.

**Strengths:**

1. The paper is clearly written and well-structured, making it easy to follow.
2. The paper provides a solid theoretical analysis.
3. The method introduces a novel approach with clear innovation.

**Weaknesses:**

1. The study is restricted to RoBERTa and LLaMA, and testing on more models would further demonstrate the generality of the approach.
2. The paper lacks a discussion of the method’s limitations, which would provide more insight into its scope and potential application.
3. The method currently aggregates sketches from different clients without explicitly considering potential conflicts between diverse update directions, which may affect convergence and global model stability.
4. Some formulas are not numbered.

**Questions:**

Please see weaknesses.

---

> ### Author Response · Authors · 2025-11-19
> **Response**
>
> We thank the reviewer for the time, effort, and valuable feedback.
>
> ### **W1. Evaluation beyond LLaMA and RoBERTa**
>
> In Section 5, we evaluated FSLoRA with RoBERTa model and GLUE benchmark and LLaMA-3.2-3B model and commonsense reasoning benchmark. In Appendix G, we further validated FSLoRA's performance with LLaMA-7B model and Wizard, Dolly-15k, and Alpaca datasets.
>
> To address this comment, we further extended our evaluation to the Owen model. We fine-tuned **Qwen2.5-1.5B-Instruct** on the commonsense reasoning benchmark using the same setup as in Table 5.1 of the original manuscript.
>
> | Method     | ARC-c | ARC-e | BoolQ | HellaSwag | OBQA  | PIQA  | SIQA  | WinoGrande | Avg.  |
> | ---------- | ----- | ----- | ----- | --------- | ----- | ----- | ----- | ---------- | ----- |
> | HeteroLoRA | 70.36 | 86.53 | 65.50 | 68.55     | 79.01 | 74.93 | 68.25 | 68.04      | 72.65 |
> | FlexLoRA   | 73.41 | 88.07 | 63.32 | 71.30     | 76.44 | 78.02 | 70.94 | 68.56      | 73.76 |
> | FLoRA      | 70.28 | 84.51 | 60.90 | 68.82     | 69.58 | 73.45 | 67.53 | 66.27      | 70.17 |
> | **FSLoRA** | 75.82 | 88.16 | 65.08 | 74.93     | 79.82 | 80.72 | 72.62 | 71.29      | 76.06 |
>
> As shown in this table, FSLoRA shows competitive performance against the heterogeneous LoRA baselines, indicating its potential to generalize beyond LLaMA and RoBERTa models.
>
> > These results, along with our earlier experiments on RoBERTa, LLaMA-3.2-3B, and LLaMA-7B, suggest that FSLoRA can be effective across different model architectures. The above table has been included in **Appendix G** of the revised manuscript.
>
> ### **W2. Discussion of the method’s limitation**
>
> We thank the reviewer for this suggestion. As with many theoretical studies [R1-R3], our work has certain limitation that we acknowledge below.
>
> A potential limitation is that our paper is primarily theoretical. Extending the proposed techniques to practical network environments and evaluating their behavior under unstable connections, delayed clients, and latency fluctuations remains an important direction for future work.
>
> > We have added a new **Limitation** section in the revised manuscript, which includes the discussion above.
>
> [R1] Cho et al., “Heterogeneous LoRA for Federated Fine-Tuning of On-Device Foundation Models,” EMNLP 2024
>
> [R2] Wang et al., “FedLoRA: Federated Fine-Tuning LLMs with Heterogeneous Low-Rank Adaptations,” NeurIPS 2024
>
> [R3] Bai et al., “Federated Fine-Tuning of LLMs under Heterogeneous Tasks and Resources,” NeurIPS 2024
>
> ### **W3. Potential conflicts between diverse update directions**
>
> **Sketching is independent across clients.** In our design, random-k sketching samples the rank-1 components $\\{b\_1 a\_1^\\top, \\dots, b\_r a\_r^\\top\\}$ **independently** for each client and for each round. This independence ensures that, in expectation,
> $$
> \\mathbb{E}\_{S}\\left[\\frac{1}{N}\\sum\_{i=1}^N B\_i S\_i A\_i\\right]= \\frac{1}{N}\\sum\_{i=1}^N B\_i \\mathbb{E}[S\_i] A\_i= \\frac{1}{N} \\sum\_{i=1}^N B\_i A\_i,
> $$
> which is **unbiased**. Introducing additional coordination mechanisms among clients may inadvertently induce bias. Furthermore, we have established the convergence of this aggregation strategy in Section 4 and validated its empirical effectiveness in Section 5 and the Appendix of the manuscript.
>
> To further support this, we report the **gradient norm** of random-k sketching and Federated LoRA (each averaged over 300 samples) at every 10 communication rounds on the LLaMA-3.2-3B with commonsense reasoning benchmark.
>
> | Rounds                        | 0     | 10     | 20     | 30     | 40     | 50     | 60     | 70     | 80     | 90     | 100    |
> | ------ | ----- | ------ | ------ | ------ | ------ | ------ | ------ | ------ | ------ | ------ | ------ |
> | FSLoRA (random-k)             | 0.312 | 0.1919 | 0.1479 | 0.0676 | 0.0729 | 0.0611 | 0.0729 | 0.0268 | 0.0193 | 0.0146 | 0.0097 |
> | Federated LoRA (no sketching) | 0.305 | 0.1881 | 0.1328 | 0.0712 | 0.0651 | 0.0583 | 0.0696 | 0.0319 | 0.0175 | 0.0129 | 0.0093 |
>
> > Overall, the gradient norms of FSLoRA and Federated LoRA follow similar decay patterns, confirming that FSLoRA preserves convergence under random-k  sketching.
>
> ### **W4. Some formulas are not numbered**
>
> Thanks for the reminder. Originally, some equations were left unnumbered because they were not referenced in the main text. **We have now added numbers to all equations for consistency.**
>
> Again, we appreciate the reviewer’s time and thoughtful comments.

---

> ### Author Response · Authors · 2025-11-27
>
> Dear Reviewer QhCG,
>
> We hope this message finds you well.
>
> We want ensure we have addressed your concerns satisfactorily. If there are any additional points you'd like us to consider, please let us know. Your feedback is important to us, and we're eager to address any remaining concerns that you may have.
>
> Thank you for your time and effort in reviewing our work.
>
> Sincerely,
>
> The authors of Submission 12821

---

### Official Review · Reviewer_65bN · 2025-10-31

**Soundness:** 4
**Presentation:** 3
**Contribution:** 3
**Rating:** 6
**Confidence:** 5

**Summary:**

This paper proposes Federated Sketching LoRA (FS-LoRA) to address resource heterogeneity in federated LoRA fine-tuning. The method maintains a high-rank global LoRA module, while clients train only a sparse subset of it, defined by a client-specific "sketching matrix". This allows the training sparsity to be adapted to each client's resources. The authors provide a convergence analysis and show that FS-LoRA avoids the overheads of prior methods, demonstrating superior performance and efficiency in extensive experiments.

**Strengths:**

- **Flexible and Intuitive Method:** The core idea of using sketching to let clients train a sparse subset of a larger global LoRA module is an elegant solution. The adaptable sketching ratio provides a flexible way to handle client-specific resource constraints.
- **Strong Empirical Results:** FS-LoRA is shown to be highly effective, consistently outperforming all SOTA heterogeneous LoRA baselines (HETLoRA, FlexLoRA, FLORA) in accuracy across multiple models and datasets.
- **Thorough Ablation Studies:** Key design choices are well-justified through comprehensive ablations, which validate the benefits of the sketching mechanism, the use of a high-rank global module, and the impact of the sketching ratio

**Weaknesses:**

- Communication cost for download: In table 3, it seems the FS-LoRA’s communication cost for download is ≥ q, but actually for methods like HETEROLoRA and FlexLoRA, the download is $k_iq/r$, if the number of rank is assume to be a public information for each client. However, q communication cost seems to be required for FS-LoRA.
- Seems sketching shares very similar intuition with sparse training, which can be another approach to solve the heterogeneous rank problem. While author has illustrated FS-LoRA’s effectiveness by theory, an empirical comparison with other sparse FL method will be strongly support FS-LoRA’s effectiveness.

**Questions:**

see weakness

---

> ### Author Response · Authors · 2025-11-19
> **Response**
>
> We thank the reviewer for the time, effort, and valuable feedback.
>
> ### **W1. Downlink communication cost of FSLoRA, HETEROLoRA, and FlexLoRA**
>
> The downlink communication is typically measured at the server side and is usually implemented as a **broadcast**, where the server sends the global model once and each client retrieves the part it needs. This broadcast-based setting is standard in FL and yields the most efficient and lowest-latency communication, which is reason that the communication cost in Table 3 is computed under this protocol. For FSLoRA, the extra cost is the point-to-point transmission of the binary sketching index.
>
> The reviewer mentioned an alternative scheme in which **each client downloads only the portion of the model it requires directly from the server**. Under this fully point-to-point strategy, the download cost (server-side) for HeteroLoRA and FlexLoRA becomes $\frac{k_i}{r} q$ per client, and the total server-side duty becomes $\frac{\sum_{i=1}^N k_i}{r} q$. However, this scheme increases the server-side workload and latency and is generally less efficient than broadcast communication. Under the same point-to-point scheme, FSLoRA also requires each client to download the sketched $\mathbf{B}$ and $\mathbf{A}$, which the server generates each round. Thus, the per-client cost is also $\frac{k_i}{r} q$, and the total server duty remains $\frac{\sum_{i=1}^N k_i}{r} q$. Therefore, in this scheme, the download cost of FSLoRA, HeteroLoRA, and FlexLoRA becomes equivalent.
>
> > The communication costs in Table 3 are reported under the broadcast setting, which is the standard, efficient protocol for disseminating the global model in FL. Under this scheme, the server sends the global model once and each client simply retrieves the portion it needs.
>
> ### **W2. Other sparse FL method for heterogeneous rank problem**
> We thank the reviewer for this insightful comment. Sketching indeed belongs to a broad class of sparse training methods. In particular, our random diagonal sketching guarantees **structural sparsity**, which is the key property that enables FSLoRA to adapt efficiently to heterogeneous LoRA settings. Beyond sketching, there exist other mainstream sparse training approaches, including **parameter sparsification** and **communication sparsification/compression** methods. **Some of them have already been considered in our manuscript, while others are not suitable for heterogeneous LoRA**. We detail these distinctions below.
>
> **Parameter sparsification**
>
> **1. Structural sparsification (baseline HeteroLoRA).** The baseline performs a structural sparsification on the LoRA by truncating fixed rows/columns of $\mathbf{B}$ and $\mathbf{A}$, then padding them back to the global size on the server. This rigid sparsification pattern introduces biased updates and lacks theoretical convergence guarantees. For clarification, we presented the performance comparison below.
>
> **GLUE benchmark (RoBERTa model)**
>
> |Method|QNLI|MRPC|CoLA|MNLI|RTE|SST-2|QQP |Avg.|
> | --| -| - | - | - | - | - | - | - |
> | Parameter sparsification (HeteroLoRA) | 87.5 | 84.4 | 75.3 | 66.3 | 69.0 | 89.5  | 85.3 | 79.6 |
> | FSLoRA  | 88.0 | 87.3 | 82.2 | 76.4 | 69.8 | 93.5  | 85.8 | 83.3 |
>
> **Commonsense reasoning benchmark (LLaMA-3.2-3B model)**
>
> | Method  | ARC-c | ARC-e | BoolQ | HellaSwag | OBQA | PIQA | SIQA | WinoGrande | Avg. |
> | - | --| -| - | - | - | - | - | - | - |
> | Parameter sparsification (HeteroLoRA) | 73.4  | 86.6  | 65.8  | 73.0      | 71.4 | 80.9 | 73.8 | 72.0       | 74.6 |
> | FSLoRA  | 76.1  | 87.2  | 69.3  | 82.2      | 80.7 | 84.0 | 76.8 | 79.1       | 79.4 |
>
> > As demonstrated in results, the deterministic sparsification used in HeteroLoRA results in inferior performance to FSLoRA.
>
> **2. Non-structural sparsification.** Another parameter sparsification scheme is to apply Top-k sparsification to LoRA parameters. However, even if parameters are made sparse, the gradient $\\nabla\_{B,A} f(\\operatorname{Top-k}(\\mathbf{B}), \\operatorname{Top-k}(\\mathbf{A}))$ remains **dense**. Therefore, clients still need to compute full gradients and perform updates in dense form, providing no computational benefit and violating the sparsity requirement for heterogeneous LoRA.
>
> **Communication sparsification.**
>
> In the original manuscript, we discussed the relationship between FSLoRA and communication sparsification/compression in Section 3.4. The key takeaway is that communication sparsification alone does **not** reduce the computation required on the client side, and therefore does not satisfy the requirements of a heterogeneous LoRA system, where both computation and communication efficiency are essential. Notably, sketching and communication sparsification are **complementary techniques** and can be combined to further improve overall efficiency. We empirically validated this synergistic effect in Appendix H.4 of the original manuscript.
>
> Again, we appreciate the reviewer’s time and thoughtful comments.

---

> ### Author Response · Authors · 2025-11-27
>
> Dear Reviewer 65bN,
>
> We hope this message finds you well.
>
> We want ensure we have addressed your concerns satisfactorily. If there are any additional points you'd like us to consider, please let us know. Your feedback is important to us, and we're eager to address any remaining concerns that you may have.
>
> Thank you for your time and effort in reviewing our work.
>
> Sincerely,
>
> The authors of Submission 12821

---

### Official Review · Reviewer_5inb · 2025-11-01

**Soundness:** 3
**Presentation:** 3
**Contribution:** 2
**Rating:** 4
**Confidence:** 4

**Summary:**

FSLoRA addresses heterogeneous client resources in federated LoRA fine-tuning by maintaining a high-rank global LoRA on the server while each client updates only a randomly selected subset of LoRA channels (“sketches”) sized to its compute and bandwidth. This preserves the accuracy benefits of a larger global rank yet makes client-side training and communication sparse and controllable; the authors provide convergence analysis showing how the sketching ratio governs the efficiency–accuracy trade-off. Empirically, FSLoRA achieves stronger results than prior heterogeneous-LoRA baselines on language understanding and reasoning tasks at comparable or lower resource cost, with negligible overhead from transmitting sketch indices.

**Strengths:**

Resource-adaptive and easy to deploy: server keeps a high-rank global LoRA while clients update sketch-selected channels, avoiding heavy SVD/merging and adding negligible downlink overhead.

Unbiasedness and convergence guarantees with the sketching ratio (k/r) cleanly controlling the accuracy–compute/communication trade-off, recovering standard FedAvg as k→r.

Robust empirical gains: consistently outperforms heterogeneous-LoRA baselines on GLUE and commonsense tasks at equal or lower cost, remaining stable with many clients and pronounced heterogeneity.

**Weaknesses:**

Random sketches can neglect useful channels over time; without smart re-sampling or scheduling, some capacity may stay undertrained. This manifests as persistent “cold” rows/columns in the LoRA factors, widening dispersion in per-channel gradient magnitudes, and a mismatch between channel salience and training frequency. The effect is amplified under non-IID data and non-stationary tasks, leading to biased adaptation where frequently sampled channels overfit client subpopulations while underexposed channels lag, degrading generalization and stability across rounds.

System realities like stragglers/partial participation aren’t deeply addressed; uneven participation could skew which channels get trained. When only a subset of clients reports in a given round, the corresponding sketched channels receive disproportionate updates, creating temporal and channel-wise imbalance. This induces drift in the effective training distribution, slows or destabilizes convergence, and can entrench performance disparities across clients whose data or resources systematically limit participation.

Baseline clarity is lacking: the paper doesn’t clearly report each model’s pre-finetuning (zero-shot or supervised) accuracy per dataset, making it hard to quantify absolute gains; without transparent “from–to” numbers by task/model, it remains ambiguous whether the observed improvements reflect real utility over strong base models or are largely within noise and tuning variance.

**Questions:**

Please see weaknesses

---

> ### Author Response · Authors · 2025-11-18
> **Responses (1/2)**
>
> We thank the reviewer for the time, effort, and valuable feedback.
> ### **W1-1. Neglect useful channels & persistent cold rows/columns**
> We appreciate the reviewer’s comments. We would like to clarify several points regarding how our random-k sketching scheme operates.
>
> LoRA parameterizes the update matrix as  $\mathbf{W} = \mathbf{BA} = b_1 a_1^\top + b_2 a_2^\top + \cdots + b_r a_r^\top,$ where $b_i$ is the $i$-th column of $\mathbf{B}$ and $a_i^\top$ is the $i$-th row of $\mathbf{A}$. In standard LoRA initialization, $\mathbf{B}$ is sampled with i.i.d. Gaussian entries and $\mathbf{A}$ is initialized to zero. This distribution is invariant under permutations of the $r$ columns of $\mathbf{B}$ and the corresponding rows of $\mathbf{A}$, so the rank-1 components $\\{b\_i a\_i^\\top\\}\_{i=1}^r$ are statistically indistinguishable and have the same expected contribution to $\mathbf{W}$.
>
> In FSLoRA, the sketch $S\\sim \\mathcal{S}\_i$ varies over iterations, and each rank-1 component $\\{b\_i a\_i^\\top\\}\_{i=1}^r$ is sampled with equal probability. Over training, this random time-varying sketches ensure that all columns and rows receive **uniform exposure** in expectation, mirroring standard LoRA.
>
> We empirically illustrate this by setting  $r=16$ and $k=4$. The table below reports the sampling frequency of each rank-1 component $\\{b\_i a\_i^\\top\\}\_{i=1}^{16}$ under two different total number of rounds:
>
> |Rounds|1|2|3|4|5|6|7|8|9|10|11|12|13|14|15|16|
> |-|-|-|-|-|-|-|-|-|-|-|-|-|-|-|-|-|
> |200|41|56|47|36|53|59|62|46|41|53|55|41|52|53|55|50|
> |400|109|110|103|105|92|100| 96 |90|100|98|100|110|95|96|107|89|
>
> > The results show that all 16 components are sampled at similar frequencies, which concentrate around the expectation as $T$ grows, **indicating that channels are unlikely to remain under-trained over time**. We further emphasized this in **Section 3.3 of the revised manuscript.**
>
> ### **W1-2. Widening dispersion in per-channel gradient magnitudes**
> As established in Theorem 4.4, random-k sketching guarantees convergence in terms of the gradient norm. The overall gradient theoretically decays over training. The unbiasedness ensures no systematic drift and mitigates the risk of widening dispersion: each channel is selected with equal probability, ensuring all channels receive sufficient updates over training. Moreover, **random sketching has a long history of theoretical support in optimization** (e.g., linear systems [R1], least-squares problems [R2]).
>
> To further support this, we report the **gradient norm** of random-k sketching and Federated LoRA (each averaged over 300 samples) at every 10 communication rounds on the LLaMA-3.2-3B with commonsense reasoning benchmark.
>
> |Rounds|0|10|20|30|40|50|60|70|80|90|100|
> |-|-|-|-|-|-|-|-|-|-|-|-|
> |FSLoRA (random-k)|0.312|0.1919|0.1479|0.0676|0.0729|0.0611|0.0729|0.0268|0.0193|0.0146|0.0097|
> |Federated LoRA (no sketching)|0.305|0.1881|0.1328|0.0712|0.0651|0.0583|0.0696|0.0319|0.0175|0.0129|0.0093|
>
> > Overall, the gradient norms of FSLoRA and Federated LoRA follow similar decay patterns, confirming that random-k sketching preserves convergence behavior.
>
> [R1] "Randomized iterative methods for linear systems." *SIAM J. Matrix Anal. Appl.* 2015.
>
> [R2] "Iterative Double Sketching for Faster Least-Squares Optimization." *ICML 2022*.
>
> ### **W1-3. Mismatch between channel salience and training frequency**
> As discussed in our response to **W1-1**, the rank-1 components $\\{b\_i a\_i^\\top\\}\_{i=1}^r$ have the same expected contribution to $\mathbf{W}$. Under random-k sketching, each column and row of the LoRA modules is sampled and updated uniformly in expectation as training progresses.
>
> It remains an open question which metric is most appropriate for quantifying the salience of each LoRA column and row (even in the centralized LoRA training).  In **Appendix C**, we investigated the spectral norm of each rank-1 component, $\\|b_i a_i^\top\\| = \\|b_i\\|\\|a_i\\|$, as a theoretically motivated proxy inspired by matrix analysis. We also evaluated a simpler alternative metric based on the magnitude sum, $\\|b_i\\| +\\|a_i\\|$. During rebuttal, we further tested an unbiased importance-aware scheme built on the spectral norm by adjusting the sketching weights, setting $s_i = \frac{1}{\pi_i}$ or 0 (not selected) and $\pi_i = \frac{\\|b_i\\|\\|a_i\\|}{\sum_{j=1}^r \\|b_i\\|\\|a_i\\|}$ , the marginal selection probability of index $i$, presented below.
>
> |Metric|ARC-c|ARC-e|BoolQ|HellaSwag|OBQA|PIQA|SIQA|WinoGrande|Avg.|
> |-|-|-|-|-|-|-|-|-|-|
> |Spectral nom|71.9|86.5|55.2|75.4|73.4|81.1|72.5|69.7|73.2|
> |Spectral nom (Unbiased )|73.4|86.2|60.5|77.9|75.8|81.6|74.6|73.1|75.4|
> | Random-$k$|75.8|86.7|69.7|81.4|80.4|83.9|76.2|78.8|79.1|
>
> > Overall, random-k remains the strongest method, despite our several attempts to improve performance through multiple potential salience-aware sampling strategies. If the reviewer has further recommendations, we are happy to explore them.

---

> ### Author Response · Authors · 2025-11-19
> **Responses (2/2)**
>
> ### **W1-4. Frequently sampled channels overfit&underexposed channels lag & Non-IID setup**
> We thank the reviewer’s comments. We would like to clarify that all channels are sampled with equal probability under random-k sketching. As shown in our response to **W1-1**, the frequencies of each component sampled will **concentrate around the expected value** as the training proceeds.
>
> Additionally, our empirical results already cover the non-IID setting: the main experiments **in Section 5** are conducted under non-IID data, and we further vary the heterogeneity level and the number of clients in **Appendix F** of the original manuscript. Across all these scenarios, FSLoRA consistently achieves higher accuracy and demonstrates stable convergence behavior, even as heterogeneity increases. These results confirm that our method's robustness under non-IID setups.
>
> > With random-k sketching, **all channel will be exposed uniformly** in expectation, as shown in the table presented in response to **W1-1**.
>
> ###  **W2. Stragglers/partial participation aren’t deeply addressed, uneven participation skew which channels get trained**
> We would like to clarify that partial participation does **not** skew which channels get trained under FSLoRA.
>
> **Sketching and client participation are independent.** Random-$k$ sketching samples the rank-1 components $\\{b\_1 a\_1^\\top, \\dots, b\_r a\_r^\\top\\}$ independently per client and per round. Each component is selected with equal probability regardless of which clients participate. Thus, uneven client participation does not bias channel selection, and all channels receive uniform exposure over training in expectation (as discussed in response to **W1-1**).
>
> **Unbiasedness remains under uniform partial participation**. Let $C$ denote the set of participating clients. Under uniform partial participation (the standard assumption in analyses of FedAvg), we have $\\mathbb{E}\_{C}\\left[\\frac{1}{|C|}\\sum\_{i \\in C} B\_i A\_i\\right]
> = \\frac{1}{N} \\sum_{i=1}^N B\_i A\_i $. Given the **independence between random-$k$ sketching and client sampling**,  we still have
> $$
> \\mathbb{E}\_{C,S}\\left[\\frac{1}{|C|}\\sum\_{i \\in C} B\_i S\_i A\_i \\right]
> = \\frac{1}{N} \\sum\_{i=1}^N B\_i \\mathbb{E}[S\_i] A\_i
> = \\frac{1}{N}\\sum\_{i=1}^N B\_i A\_i,
> $$
> which remains **unbiased**. The effect of partial participation on FSLoRA mirrors its effect on FedAvg. Extending the theory from full to partial participation follows standard FL techniques [R1, R2].
>
> **Empirical evidence.** In Appendix F, we include partial-participation experiments where only 10 out of 50 or 100 clients participate in each round. As reported in **Tables F1–F2 and Figure 4**, FSLoRA demonstrates its stable convergence and maintains its advantage over baselines. For clarification, we present Table F2 below, where 10 clients participate in each round and the total number of clients is 100.
>
> |Method|ARC-c|ARC-e|BoolQ|HellaSwag|OBQA|PIQA|SIQA|WinoGrande|Avg.|
> |-|-|-|-|-|-|-|-|-|-|
> | HeteroLoRA|71.76|86.24|62.57|68.07|76.60|79.38|74.10|69.69|73.55|
> | FlexLoRA | 73.38 | 87.54 | 69.03 | 75.27 | 78.60 | 80.47 | 74.16 | 73.80 |76.53|
> | FLoRA | 69.97 | 83.25 | 67.10 | 71.67 |73.60 | 78.94 | 72.21 | 70.80 | 73.44 |
> | FSLoRA| 74.40 | 87.54|70.13 |79.90|79.40 | 83.57 | 76.51 | 78.93| 78.80 |
>
> > **Stragglers and partial participation is not unique to FSLoRA,** which affect all FL methods. Our theoretical analysis focuses on the full-participation setting to **isolate and clearly highlight the impact of the sketching mechanism**.
>
> [R1] "Achieving linear speedup with partial worker participation in non-iid federated learning." *ICLR 2021*.
>
> [R2] "Tackling the objective inconsistency problem in heterogeneous federated optimization." *NeurIPS 2020*.
>
> ### **W3. Report each model’s pre-finetuning accuracy**
>
> In Figure 6 of the original manuscript, we plotted the accuracy curve for each task as training progresses. The results show a clear upward trend, indicating stable improvement. To further address this comment, we include the zero-shot performance of the base model prior to fine-tuning. The performance of FSLoRA is drawn from Table 5.1 of the original manuscript.
>
> **GLUE benchmark (RoBERTa model)**
> | Method| QNLI | MRPC | CoLA | MNLI | RTE| SST-2 | QQP | Avg.|
> |-|-|-|-|-|-|-|-|-|
> | Pre-finetuning | 49.5 | 52.0 | 30.9 | 32.6 | 47.3 | 50.9  | 63.2 | 46.6 |
> | FSLoRA | 88.0 | 87.3 | 82.2 | 76.4 | 69.8 | 93.5  | 85.8 | 83.3 |
>
> **Commonsense reasoning benchmark (LLaMA-3.2-3B model)**
> | Method| ARC-c | ARC-e | BoolQ | HellaSwag | OBQA | PIQA | SIQA | WinoGrande | Avg. |
> |-|-|-|-|-|-|-|-|-|-|
> | Pre-finetuning | 51.5  | 61.4  | 47.2  | 55.8 | 52.4 | 21.3 | 54.4 | 4.6 | 43.6 |
> | FSLoRA | 76.1  | 87.2  | 69.3  | 82.2 | 80.7 | 84.0 | 76.8 | 79.1 | 79.4 |
>
> > Overall, FSLoRA delivers clear and substantial gains over the base model.
>
> Again, we appreciate the reviewer’s time and thoughtful comments.

---

> ### Author Response · Authors · 2025-11-27
>
> Dear Reviewer 5inb,
>
> We hope this message finds you well.
>
> We want ensure we have addressed your concerns satisfactorily. If there are any additional points you'd like us to consider, please let us know. Your feedback is important to us, and we're eager to address any remaining concerns that you may have.
>
> Thank you for your time and effort in reviewing our work.
>
> Sincerely,
>
> The authors of Submission 12821

---

### Author Response · Authors · 2025-12-02
**Summary**

Dear Area Chair,

We greatly appreciate your time and effort in handling our submission. In this work, we propose federated sketching LoRA (FSLoRA), which leverages a sketching mechanism to enable clients to update only random submatrices of global LoRA modules maintained by the server each round. By adjusting the sketching ratios, FSLoRA flexibly adapts to client-specific communication and computational constraints, effectively catering to resource heterogeneous systems. We provide a rigorous convergence analysis that characterizes how the sketching ratios affect the convergence of FSLoRA. Through comprehensive experiments on multiple datasets and LLM models, we demonstrate that FSLoRA achieves consistent performance improvements over various baselines.

At this stage, we would like to offer a brief summary of the reviewers' main concerns and how we addressed them in our rebuttal.

**Reviewers 65bN and QhCG** took a positive stance and requested only a few additional experiments and clarifications, which we have provided in the rebuttal.

**Reviewers 5inb and E2kV** raised the following concerns that were largely driven by misunderstandings of our **sketching mechanism**, which we clarified thoroughly during the rebuttal phase.

1. *Random-k sketching may neglect/discard useful channels* (Reviewer 5inb and E2kV)

2. *Uneven participation could skew which channels get trained* (Reviewer 5inb)

3. *FSLoRA introduces extra server-side overhead* (Reviewer E2kV)

In our response, we clarified that:

1. Random-k sketching **does not** bias learning toward or against any LoRA row/column. It exposes all LoRA rows and columns uniformly in expectation as training proceeds, a property ensured by the law of large numbers. The reviewers’ concern arises from a misunderstanding of this core property. We added numerical experiments to illustrate this in our rebuttal. Moreover, we explored several alternative strategies, but we did not observe advantages over random-k sketching. These results are presented in Appendix C.

2. Partial participation **does not** skew which channels get trained, since sketching and client participation are independent. As shown in Appendix F, FSLoRA converges properly under partial participation.

3. FSLoRA introduces **almost minimal** server-side overhead compared to heterogeneous LoRA baselines. We provided both a detailed server-side complexity analysis (also included in Appendix A) and a numerical per-round aggregation latency comparison in the response.

Although the reviewers did not follow up after their initial reviews, we believe their concerns were addressed in our rebuttal.

Thank you again for your coordination.

Best regards,

Authors

---

### Meta-Review · Area_Chair_4cPG · 2026-01-04

**Summary:**

This paper proposed sketching LoRA for fine-tuning LLMs in federated learning. This paper received borderline scores (4, 4, 6, 6). Unfortunately the reviewer engagement during discussion is low, and there are no signs of score changes.

Reviewers raised several clarification questions that are reasonably addressed in the authors' responses. However, concerns on sketching efficiency and cost are not fully addressed. Sketching for communication compression is well studied for both theory and practice, but sketching for model training is less studied. Like reviewers alluded to, the random selection in sketching often leads to a trade-off in convergence speed and compression ratio, and neglects the structure in neural networks and matrix decomposition. I would encourage the authors to revisit the contributions in both theory and practice, and highlight them in the rich literature of LoRA and parameter selection in federated learning.

**Reviewer Concerns:**

Reviewers raised several clarification questions that are reasonably addressed in the authors' responses. However, concerns on sketching efficiency and cost are not fully addressed. Sketching for communication compression is well studied for both theory and practice, but sketching for model training is less studied. Like reviewers alluded to, the random selection in sketching often leads to a trade-off in convergence speed and compression ratio, and neglects the structure in neural networks and matrix decomposition.

**Reviewer Scores:**

This paper received borderline scores (4, 4, 6, 6). Unfortunately the reviewer engagement during discussion is low, and there are no signs of score changes. After considering the authors' comments, this paper will receive borderline scores.

---

### Decision · Program_Chairs · 2026-01-26

Reject